# Contribution of the Golgi apparatus in morphogenesis of a virus-induced cytopathic vacuolar system

Ranjan Sengupta[1,2,3] , Elaine M Mihelc[2], Stephanie Angel[1,2,3], Jason K Lanman[2], Richard J Kuhn[2,3] , Robert V Stahelin[1,3]

The Golgi apparatus (GA) in mammalian cells is pericentrosomally anchored and exhibits a stacked architecture. During infections by members of the alphavirus genus, the host cell GA is thought to give rise to distinct mobile pleomorphic vacuoles known as CPV-II (cytopathic vesicle-II) via unknown morphological steps. To dissect this, we adopted a phased electron tomography approach to image multiple overlapping volumes of a cell infected with Venezuelan equine encephalitis virus (VEEV) and complemented it with localization of a peroxidase-tagged Golgi marker. Analysis of the tomograms revealed a pattern of progressive cisternal bending into double-lamellar vesicles as a central process underpinning the biogenesis and the morphological complexity of this vacuolar system. Here, we propose a model for the conversion of the GA to CPV-II that reveals a unique pathway of intracellular virus envelopment. Our results have implications for alphavirus-induced displacement of Golgi cisternae to the plasma membrane to aid viral egress operating late in the infection cycle.

## Introduction

In eukaryotic cells, membrane-bound organelles transition over their lifetime through multiple morpho-functional states (Heald & Cohen-Fix, 2014; Westrate et al, 2015). In addition to endogenous events, morphological remodeling of organelles is also observed during virus infection of the host cell. Viruses have evolved to exploit host cell structural and functional plasticity of organelles and thus are important tools to dissect and understand cellular trafficking pathways (Welsch et al, 2009; Den Boon et al, 2010; Hsu et al, 2010; Hollinshead et al, 2012; Romero-Brey et al, 2012; Chen et al, 2015; Cortese et al, 2017). Indeed, a major restructuring of the host cell secretory system has been associated with virus maturation and

egress, leading to either delayed or little consequences on cellular secretion (Mousnier et al, 2014), an indication that these pathogens operate within the critical structural and functional parameters of host organelle function. These recurrent observations necessitate the investigation of pathogen induced structural states of host organelles to obtain clues on morpho-functional states observed during viral infection.

The genus alphavirus (from the Togaviridae family) are enveloped, icosahedral (~700 Å diameter), positive-strand RNA viruses that include human pathogens such as Chikungunya virus (CHIKV) and Venezuelan equine encephalitis virus (VEEV). Alphavirus genomic RNA is packaged by capsid protein to form a spherical nucleocapsid core (NC) (~40 nm in diameter) (Jose et al, 2009; Zhang et al, 2011). Typically, the NC obtains its envelope via budding at the plasma membrane (PM) facilitated by the interaction of the membrane embedded envelope glycoproteins E1/E2 and capsid (Brown et al, 2018). The glycoproteins assemble in the ER, undergo maturation in the Golgi apparatus (GA), and are then transported to PM where they interact with capsid to initiate budding (Martín et al, 2009). However, during mid to late stages of the alphavirus infection cycle, a structurally distinct pleomorphic vacuolar system (cytopathic vesicles II or CPV-II) predominates in the cytoplasm of the host cell. CPV-II are known to originate from the GA and exhibit a characteristic accumulation of nucleocapsid core (NC) on its membrane (Mussgay & Weibel, 1962; Grimley et al, 1968). These early studies revealed the morphological complexity of CPV-II where both uni- and bilamellar forms of CPV-II were observed in the milieu. Griffiths et al (1983) demonstrated that monensin blocked E1/E2 glycoprotein transport out of the GA and resulted in the swelling and vacuolization of Golgi cisterna that bound NC cores (much like the CPV-II) as a result of accumulation of viral glycoprotein in the GA (Griffiths et al, 1983). Although this mechanism could explain biogenesis of the unilamellar CPV-II, it does not account for other forms of CPV-II, especially the bilamellar form reported in the early studies. In addition, from a cell biology perspective, neither structural nor the functional consequence of this vacuolar biogenesis on the cell secretory system is exactly known.

[1]Department of Medicinal Chemistry and Molecular Pharmacology, Purdue University, West Lafayette, IN, USA   [2]Department of Biological Sciences, Purdue University, West Lafayette, IN, USA   [3]The Purdue Institute of Inflammation, Immunology and Infectious Disease, Purdue University, West Lafayette, IN, USA

Correspondence: rstaheli@purdue.edu; rsen.nih@gmail.com
Ranjan Sengupta's present address is Angiex Inc., Cambridge, MA, USA.
Elaine M Mihelc's present address is Department of Systems Pharmacology and Translational Therapeutics, Perelman School of Medicine, University of Pennsylvania, Philadelphia, PA, USA.

The structure of the GA is known to be intrinsically connected to its various functions. During interphase, the mammalian GA is pericentrosomally anchored and exhibits a stacked structure comprised of a functionally defined set of cisternae (namely, cis-, medial-, and trans) (Klumperman, 2011; Yadav & Linstedt, 2011). However, at instances such as cell division, during certain neurodegenerative conditions and bacterial and viral infections, the GA loses its typical stacked architecture and its pericentrosomal localization (Gahmberg et al, 1986; Campadelli et al, 1993; Lavi et al, 1996; Nakagomi et al, 2008; Heuer et al, 2009; Quiner & Jackson, 2010; Corda et al, 2012; Thayer et al, 2013; Joshi et al, 2014; Kim & Satchell, 2016; Hansen et al, 2017; Wei & Seemann, 2017). Though critical to its function, a clear morphological understanding of the GA membrane remodeling pathways and their exact functional relevance is still lacking. Thus, such instances of virus induced morphological remodeling of the GA could provide excellent opportunities for studying the morpho-functional plasticity of this secretory organelle.

Here, we employed several transmission electron microscopy (TEM) approaches to elucidate host cell GA conversion into various forms of CPV-II during infection of TC-83, a vaccine strain of VEEV. Using traditional thin-section TEM, we established a timeline of structural changes of the GA during infection. We then applied an improvised hybrid sample fixation method in conjunction with peroxidase tagging of a GA marker to show that previously identified forms of CPV-II carry this marker. To gain a 3D understanding of the process, we established a phased data collection approach and took advantage of the stability of resin sections under the electron beam to collect multiple tilt-series from the same area of the cell over time. The vast array of 3D data collected enabled statistical analyses that yielded a 3D classification of the CPV-II system, identification of large and complex intermediates that links the various forms of CPV-II, the relative spatial frequency of the various forms, and a model of their morphogenesis. Thus, our data identify a GA-associated pathway of CPV-II biogenesis and proposes a model for the restructuring of Golgi cisternae into large single- and double-lamellar CPV-II forms identified in VEEV infection.

# Results

To carry out this work, we chose to use the live-attenuated vaccine strain of VEEV, TC-83. TC-83 can be handled in BSL-2, making it possible to use live infected cells for high-pressure freezing (stationed within BSL-2 lab) to provide high quality membrane preservation required for such studies. We ruled out the possibility that the critical mutation present on E2 (T120R) could influence the binding of NC to E2, the focus of this study, by modeling the mutation to the local structure of E2 (Fig S1).

### VEEV-infected cells show progressive remodeling of the GA with and emergence of pleomorphic cytopathic vesicles

A considerable GA remodeling precedes the onset of CPV-II. Golgi structures in BHK cells were screened at different time points starting at 3 h postinfection (PI) up to 12 h (Fig S2A–E). These features on the GA were absent in uninfected BHK cells (Fig S3A and C

and zoomed in Fig S3B and D red arrows). The GA architecture at 12 h PI exhibited extensive herniations and vacuolization in addition to unstacking and bending (Fig S2B, i–iv). 250-nm serial sections were screened for viewing large complex structures (Fig S2C, i, sections 1 through 20). The pleomorphic nature of the CPV-II were more evident in these thicker sections as oblong and dumbbell-shaped structures studded with NC were readily observed. Multiple CPV-II clusters were observed 12 h PI near the PM that were composed of various morphological forms of CPV-II (Fig S2E, i–iii). At this stage, CPV-II exhibited a more scattered distribution in the cytoplasm.

To detect the distribution of both early and late vacuoles (CPV-II) arising from the GA, we carried out HRP-tagging based localization of the Golgi marker α-mannosidase-II. Here, we employed a recently published hybrid method for superior sample preservation to localize HRP-tagged α-mannosidase-II (Golgi FLIPPER, Kuipers et al, 2015), a cis-medial Golgi marker (Sengupta et al, 2019). BHK cells first transfected with ManII-HRP encoding plasmid were then infected with TC-83 virus (MOI of 20) and then fixed at 6- and 12-h PI.

In control cells, a tight perinuclear staining was observed. At a higher magnification, the perinuclear stain resolved into multiple stacks of GA (Fig 1A ii and iii, blue arrowheads and area within the blue box) (Fig 1A i, region demarcated with the red box). In contrast, infected samples from 6 h PI showed stained vesicular structures scattered predominantly in the perinuclear region (Fig 1B i, region demarcated with the yellow box). At higher magnifications they resolved into multiple stained Golgi stacks (Fig 1B iii–v, blue arrowheads) and large vesicles (~200–500 nm) associated with the stacks (Fig 1B iii, yellow arrows). Furthermore, large cisternal herniations gave rise to distended structures that were still connected to the Golgi stack (white and magenta arrows, Fig 1B iii–v). Distended structures originated from both the outer periphery of the stacks (Fig 1B iii and iv, white arrows) and from the middle of the stack (magenta arrows, Fig 1B iv and v). A limited number of isolated vesicles carrying the ManII-HRP marker were detected near the PM (Fig 1B vi, green arrows). The samples imaged 12 h PI revealed CPV-II carrying the ManII-HRP stain at a high concentration not only at the perinuclear regions but also scattered throughout the cytoplasm and near the PM (area demarcated by orange box in Figs 1C i and S3A i–vi). Images of the representative section show stained vesicular structures at the perinuclear region (orange box in Fig 1C i). Images of this region at a high magnification exhibit clusters of CPV-II (Fig 1C ii, denoted by red arrows) and a mix of various morphological forms (Fig 1C iii, indicated with roman numerals I, II or IV). A dense complement of NC was observed associated with the cytopathic vesicles in these clusters.

In addition, to determine if this Golgi vacuolization is cell type–specific, HeLa cells stably expressing ManII-HRP were infected with TC-83 (Fig S4A and B). Results from the screen in HeLa cells at 6 h PI (Fig S4B) show similar features observed in BHK cells. As with BHK cells, although mock infected cell lines show tight perinuclear localized Golgi stacks (Fig S3B), infected cells exhibit Golgi stacks that were restricted mainly at the perinuclear region with cisternal herniations and vacuolization (Fig S4B iv–vi). This validation from the ManII-HRP-HeLa cell line also rules out the possibility of artefactual localization of the marker and perturbed morphology of the GA as a result of overexpression.

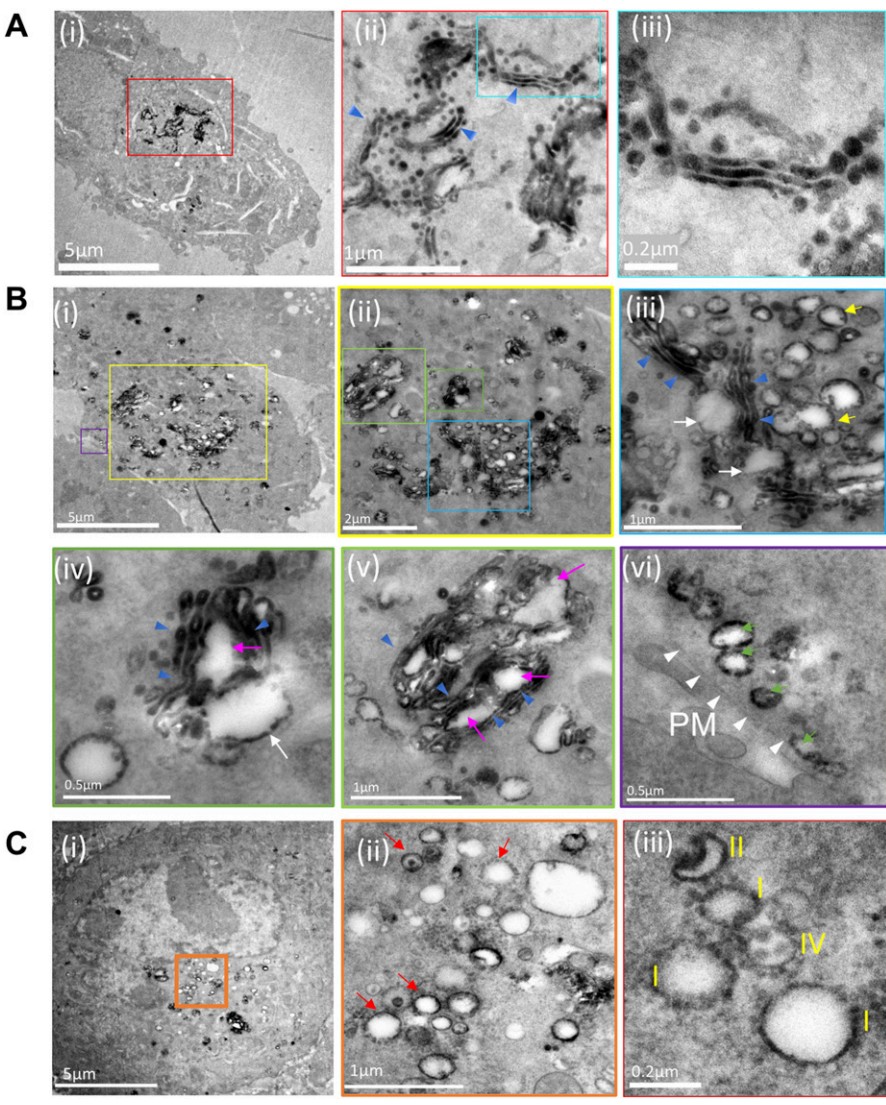

**Figure 1. Pleomorphic CPV-II in the perinuclear region and near the plasma membrane are of Golgi origin.**

**(A, B, C)** A hybrid chemical-cryo fixation method was optimized with HRP-DAB assay for the detection of HRP-tagged α-mannosidase-II, a (cis-medial) Golgi membrane marker in mock infected and in VEEV-infected BHK cells at 6 and 12 h postinfection (PI) ((A) (i)–(iii), (B) (i)–(vi), and (C) (i)–(iii), respectively). **(A)** Transmission electron microscopy images of 90-nm resin sections of mock controls show a tight perinuclear staining ((A) (i), area within the red box) that at higher magnification resolves into multiple canonical Golgi stacks ((A) (ii)). **(A)** Magnified view of the specifically labeled single stack exhibit well-preserved cis-medial stacks ((A) (iii)) area magnified within the blue box in (ii). **(B)** Cells from 6 h PI show the manII-HRP marker scattered mostly at the perinuclear region with limited punctae outside this region (B) (i) Successive magnified views ((B) (ii) through (v)) of the perinuclear region ((B) (ii)) demarcated with a yellow box in (i) and color-coded henceforth show herniated cisternae within the existing Golgi stacks ((B) (ii)–(iv), blue arrowheads), vacuolization (white arrows, (iii)–(v)), and the area around littered with vesicles of similar sizes. **(B)** These herniations were not restricted either to the *trans-* or the *cis*-face but were seen to occur from cisternae in the middle of the stacks (white arrows, (B) (iii)–(v)). **(B)** Progressive conversion of the cisternae into large vesicles and the replacement of a stacked structure with a cluster of vesicles was apparent ((B) (iv)). **(B)** These Golgi-derived vesicles ((B) (vi), yellow arrows) were also found in limited numbers at the plasma membrane ((vi), demarcated with white arrows). **(C)** Images of cells sectioned from 12-h PI exhibit different classes of CPV-II ((C) (i), demarcated with an orange box) and show the absence of stacked Golgi structures but a predominance of CPV-II with detectable nucleocapsid cores (NC) associated with the vesicles (ii, red arrows). Some NCs were also detected in free space within these clusters (iv, white arrowheads), but it could not be concluded if they were freely occurring in the cytoplasm or were attached to CPV-II excluded from the thin section. **(C)** The most prominent classes of vesicular structure observed were (1) a crescent shaped structure with higher density toward the inner curved side indicating the presence of NC, (2) a typical round vesicular structure with NC on the cytoplasmic surface, (3) a vesicle with enveloped virus within and NC on the outer surface ((C) (ii) and (iii)). See (Fig S4) for distribution of HRP-ManII carrying vesicles at the same time point and a comparison with HeLa cells stably expressing HRP-ManII.

## Serial-section tomographic reconstruction of a representative infected cell

To visualize the structural changes in the structurally complex GA, a combination of serial sectioning and image montaging was employed to obtain large-volume tomograms. First, 250-nm thick serial sections of the samples were collected and screened for cells containing multiple Golgi stacks exhibiting a structural flux and a presence of abundant CPV-II. A suitable area in a typical representative cell was then identified (Fig S5A), the volume of interest within which spanned several consecutive sections on the EM grid (see the Materials and Methods section for details). This first volume (tomogram 1, area outlined in cyan in Figs 2B and S5B [see also Video 1, segmented tomogram]) consisted of three well-defined Golgi stacks forming the basis of our GA study. Segmentation of this reconstructed volume consisting of the three Golgi stacks revealed a pathway of Golgi conversion into a vesicular system (Fig 2C, Golgi stacks in green and CPV-II in gold). Reconstruction and segmentation of tomogram 1 provided clues that led us to the collection of tilt-series for tomogram 2 (Figs S5C and D and 6E(b)) and tomogram 3 (later highlighted in Fig 8) from the same cell.

## Outward bending of Golgi cisternae that bind viral NC provide clues for the biogenesis of double-lamellar CPV-II

The most prominent feature of VEEV induced Golgi remodeling as seen in conventional 2D TEM studies is the herniation of cisternae and formation of large vacuoles (~0.2–1.5 μm) that may separate out to form single-lamellar CPV-IIs (Fig S2). However, these studies provide little information on the morphogenesis of other classes of CPV-II that were identified in our Golgi marker study (Fig 1). Thus, we

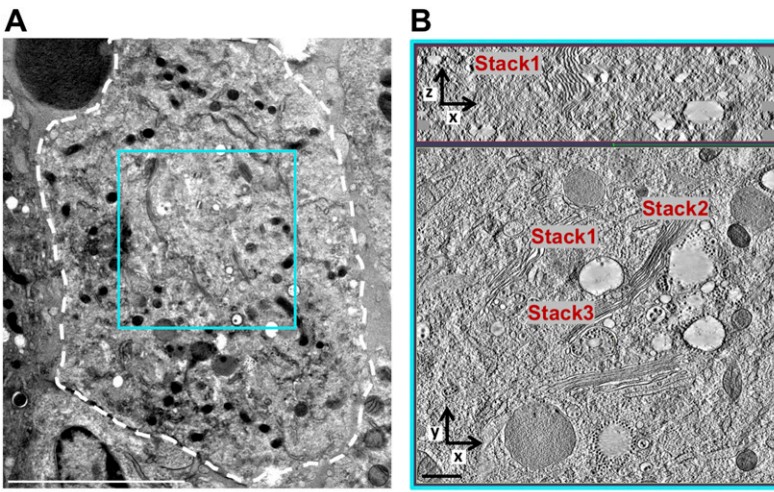

**A** 

**B** Stack1 / Stack1 / Stack2 / Stack3

**Figure 2. Large-volume tomogram of an VEEV-infected cell.**
**(A)** Transmission electron microscopy image collected at 300 kV of a 250-nm thick section of a TC-83–infected BHK cell (outlined in white). The approximate area of this cell from where 8–10 serial sections were used to collect and reconstruct tomogram 1 is demarcated by the cyan box. **(B)** A 3-nm thick virtual section from tomogram 1 (See also Video 1) with the z-depth along the x- and y-axis shown along the top and right side, respectively. The depicted slice exhibited three Golgi stacks that are denoted by stack 1, 2, and 3, respectively. **(C)** 3D visualization of the tomogram volume employing manual segmentation of Golgi (green) and nascent CPV-II Golgi (gold). Scale bars: (A) 5 µm, (B) 500 nm, and (C) 2 µm.

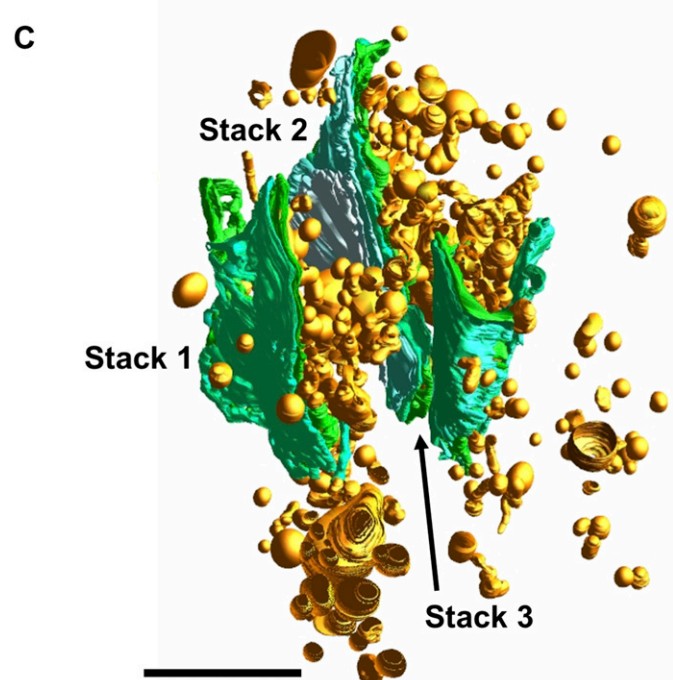

**C** 

Stack 2 / Stack 1 / Stack 3

looked for subtle structural clues on the segmented tomogram of the GA to address this. The two Golgi stacks under study (stack 1 and stack 2, Fig 3A, B, E, and F) both show the presence of viral NC on the first intact cisternal face. Many of these NCs were bound to the cisternal rims (edges) as illustrated for stack 1 and stack 2. The edges displayed curvature away from the plane of the stack at the locations where the NCs were bound (Fig 3A–H and Video 2). Close examination of the bent edges of the cisternae revealed that the section bent around NCs largely retained the narrow intralumenal distance of cisternae present in a Golgi stack (Fig 3I–L). A large curvature of the cisternal rims was composed of multiple bending events occurring around bound NC (Fig 3M–P). However, not all bound NCs were associated with cisternal bending. This was true especially with ones in the middle of the stack. To visually clarify membrane bending, we chose to highlight any bending that was

possibly associated with cisternal wrapping around the NC by selecting NCs located on membrane which had a local radius of curvature between 20 and 60 nm, corresponding to the 40 nm radius of a NC. These NCs were colored red in the 3D model (Figs 3 and S6A–C). NCs located on membranes with radius of curvature greater than 60 nm were colored blue (Fig 3).

In addition to curling of the cisternal rims, a single large cup-like transformation of a Golgi cisterna was observed (Fig 3Q–S and Video 3). This formation consisted of a largely intact cisternal structure curving away from the plane of the Golgi stack (in light green, Fig 3Q and R), with one edge closing to form a vesicle-like structure (top-down view in Fig 3S). This form essentially represents a transitional intermediate between a flat Golgi cisterna and a double-membrane vesicle. A high concentration of NCs is present on the inner membrane of this intermediate, but only 14 of 97 NCs show membrane curvature indicative

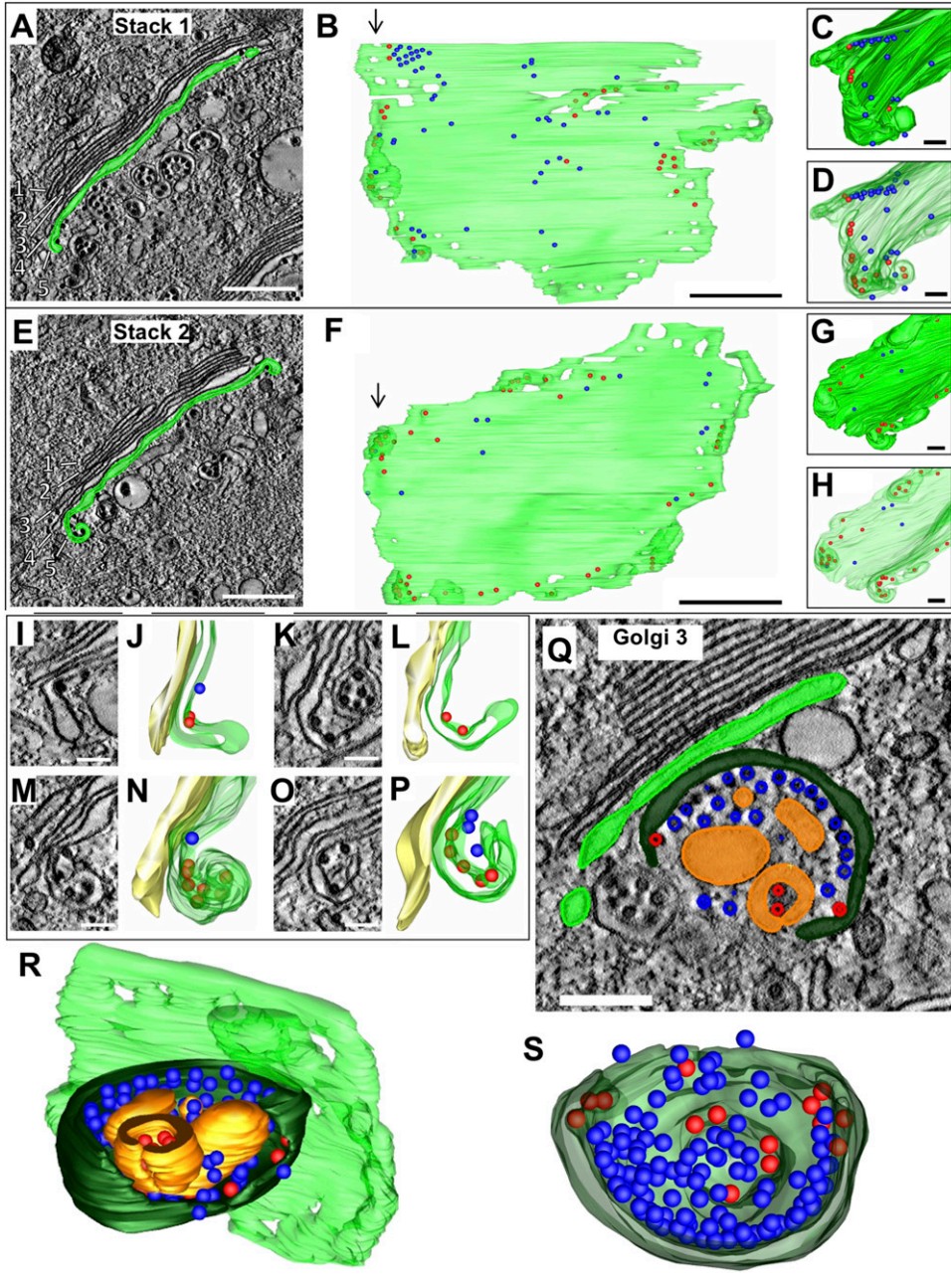

**Figure 3. Golgi cisternal rims exhibit bending in regions in contact with NC.**
**(A)** 3-nm thick virtual section from tomogram 1, showing a Golgi stack ("Stack 1") with the final intact cisterna colored green. Five cisternae are visible. **(B)** 3D rendering of the cisterna through the entire 2.5 µm depth of the reconstructed volume. Colored spheres represent NCs in contact with the Golgi membrane (colored blue or red based on the radius of curvature on the adjacent membrane as shown in Fig S6, see also Video 2). **(B, C, D)** The curled edge of the cisterna viewed from the top down of the cisternae in panel (B) (direction of arrows in (B) indicates view for (C, D)). **(A, B, C, D, E, F, G, H)** Similar views as (A, B, C, D) of a second Golgi cisterna through the depth of tomogram 1. **(I, J, K, L, M, N, O, P)** Edges of the cisternae (green) exhibit various degrees of curling away from the next cisterna in the stack (yellow). **(Q, R, S)** A third Golgi stack with a large portion of a cisterna (dark green) bending away from the adjacent intact cisterna (light green). The tomographic slice (Q) and 3D renderings (R, S) of this cupped cisterna which is bound with NC (red and blue) and encloses a double-membrane CPV-II (gold). See Video 3. Scale bars: 500 nm (A, E), 100 nm (I, M, K, O), and 200 nm (Q).

of a potential membrane wrapping event (red spheres in Fig 3S). This intermediate largely retains the characteristic intralumenal distance of a Golgi cisterna (15–19 nm) and provides the most telling evidence for the morphogenesis of double-lamellar CPV-II.

## Pleomorphic membrane structures with bound NC are fragmented Golgi cisternae

Immediately adjacent to the *trans*-face of stack 1, large pleomorphic membrane bound structures were observed (segmented tomogram in Fig 4A and Video 4). The structures closest to the Golgi stacks appeared to be discrete structures in single slices through the tomographic volume (Fig 4B, pink and brown), but 3D segmentation revealed that they are in fact partially intact Golgi cisternae still somewhat aligned with the stacked cisternae (Fig 4C and D). The central part of the structure shown in Fig 4C (area containing bound blue NC) retained flat cisternal morphology and maintained contact with the preceding cisterna in the stack (Fig 4B, arrowhead), whereas the outer edges exhibited a fenestrated morphology (Fig 4C, arrowheads). Two small double-lamellar vesicular structures surrounding small groupings of NCs were also present (Fig 4C, inset). The next closest cisternal structure, in brown, exhibited more vesicular structure than the previous, with long tubular elements connecting the vesicular formations (Fig 4D).

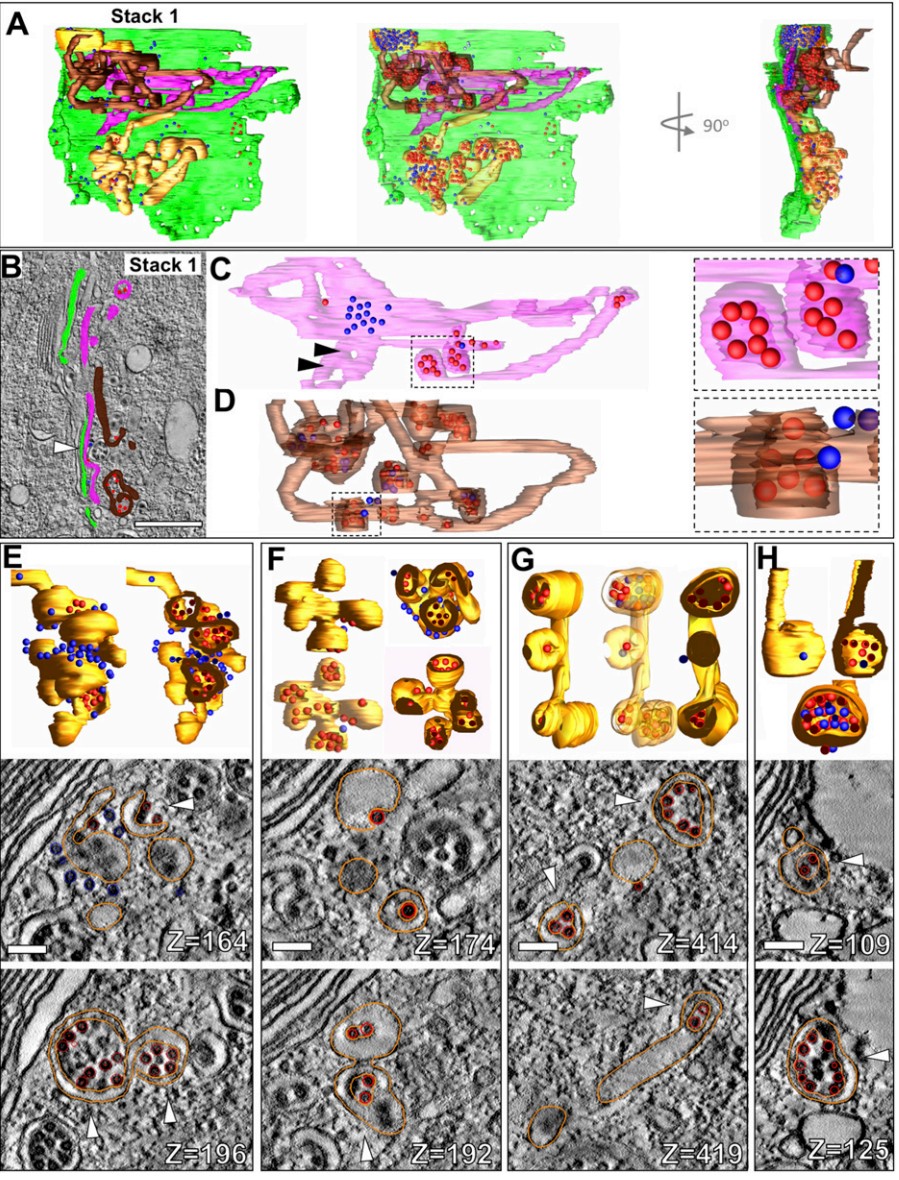

**Figure 4. Pleomorphic membrane structures with bound NC are fragmented Golgi cisternae.**
**(A)** 3D rendering of Golgi 1 *trans*-most cisterna along with selected pleomorphic CPV-II forms (gold) closest to cisternae (pink and brown) and associated NC (blue and red spheres). **(B)** 3-nm virtual section through stack 1, highlighting the intact cisterna (light green) and two large, fragmented cisternae (magenta and brown) and their associated NCs (blue and red spheres). **(B, C, D)** 3D renderings of two pleomorphic cisternal forms which are highlighted in magenta (C) and brown (D) in (B). Insets of (C) and (D) show double-lamellar nature of vesicular structures, with NC inside (red) and outside (blue). **(E, F, G, H)** CPV-II structures of varying pleomorphic shapes include forms with several double-lamellar vesicular formations (E, F) and other vesicular and tubular elements (G, H). Cutaway views in (E, F, G, H) reveal the double-lamellar nature of the vesicular forms. 3 nm thick virtual sections from two different depths of each form shown in (E) though (H) are presented with the segmentation of the structures of interest shown in gold and NC in red. The double-lamellar nature of the forms (similar to classes 2 and 3) is indicated by white arrowheads. See Video 4. Scale bars: 500 nm (B) and 100 nm (E, F, G, H).

Moving further away from the *trans*-face, pleomorphic CPV-II structures were observed to contain a combination of small vesicular formations and tubules with highly varied morphology (Fig 4E–H). Some of these structures included up to 10 connected vesicular formations, resulting in structures resembling grape bunches (Fig 4E and F). Other structures consisted of a few vesicles with tubular connections (Fig 4G), or even simpler, a single vesicle with a tubular component (Fig 4H). Tomography proved to be the key to unraveling their complex morphologies and connections (see corresponding 2D slices in Fig 4E–H).

### 3D analysis of Golgi stacks associated with large unilamellar CPV-II

One of the Golgi stacks had closely associated large vesicles, reminiscent of those observed at earlier time points by 2D EM and shown to be of GA origin (Fig 5A, compare with Fig S3A(v) and B(vi)). Although the presence of NCs on the vesicles in 2D was either not observed (at early time points) or perhaps went undetected, the large vesicles in the tomogram clearly have large numbers of NCs bound to their cytoplasmic surfaces, indicating that they are indeed CPV-II. Particularly, a large irregularly shaped vesicle of ~1.5 µm diameter in close apposition with the *trans*-most Golgi cisterna of the stack (Fig 5B and C). This vesicle was separated from the cisterna by a single layer of NC which is bound to both the CPV-II and the Golgi membrane in a bivalent association (Fig 5C–E). Segmentation and 3D visualization of this vesicle and cisterna revealed increased NC density in the region where the vesicle was bound (Fig 5F and G).

The visible deformation on the Golgi cisterna where the CPV-II was positioned further confirmed the tight association of the CPV-II with the Golgi (Fig 5G). Examination of the region immediately

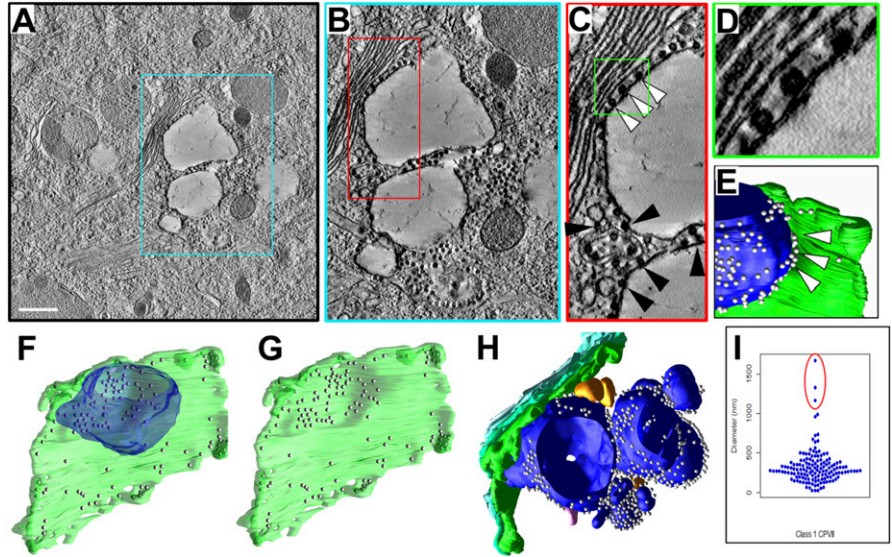

**Figure 5. 3D analysis of Golgi-associated vacuoles.**
**(A, B, C, D)** A 3-nm thick virtual section from tomogram 1 showing a large CPV-II abutting the intact *trans*-most cisterna of a Golgi stack, with increasing magnification of the region of interaction presented in the color-coded outlined areas in (A, B, C, D). **(C, D)** NCs can be observed contacting both the intact cisternal membrane and the vesicular membrane (C, white arrowheads and D), as well as between multiple CPV-II in the cluster (C, black arrowheads). **(C, D, E)** A 3D view of the segmented CPV-II region shown in (C, D) (blue), with NC (white spheres) filling much of the gap between the vesicle and the Golgi (green), indicated by arrowheads. **(A, B, C, D, E, F, G)** A 3D view depicting the entire *trans*-most cisternae shown in (A, B, C, D, E) with (F) or without (G) the large CPV-II. A concentration of NC is seen on the Golgi at the interface with the CPV-II and an indentation in the GA where the vesicle was located. **(F, H)** 3D segmented view of the cluster of CPV-II surrounding the large vesicle depicted in (F), consisting of variably sized class 1 CPV-II (blue) and their associated NCs. A few class 3 CPV-II (gold), all in close physical proximity with the Golgi stack (green shades) are also shown. **(I)** The three largest class 1 CPV-II in the volume are found in this cluster, as indicated on the plot of vesicle diameters for this class (I, circled in red). See Video 5. Scale bar: 500 nm (A).

surrounding these large vesicles revealed several additional, large, and irregularly shaped CPV-II, as well as closely associated smaller CPV-II (Fig 5H and Video 5). Vesicles were apparently connected into a large cluster by NC bivalently bound to two vesicles (Fig 5C black arrowheads). Although most of the vesicles in the cluster fell within the typical size range of CPV-IIs, it is noteworthy that the three large vesicles in this cluster were the three largest CPV-IIs found in the whole tomographic volume but distinct outliers in terms of size (Fig 5I, circled data points). The vesicle highlighted in Fig 5F and found in closest apposition with the GA is the largest CPV-II of the 353 analyzed in this volume. From our observations from the marker study and these rare large CPV-II in the tomogram, it is tempting to interpret these large CPV-II as portions of herniated Golgi cisternae. It is unknown whether these large irregularly shaped vesicles persist or are unstable intermediates that may break down into the smaller, more spherical shaped class I CPV-II observed throughout the volume.

### Identification and 3D analysis of four morphological forms of CPV-II

To date, published ultrastructural data on CPV-II has been obtained via traditional thin section studies. Due to the complex morphological nature of CPV-II, it is difficult to ascertain its real 3D form by thin-section screening. In addition, because Golgi are an architecturally complex organelle, we undertook a comprehensive 3D study on CPV-IIs (utilizing 3D data from two overlapping tomograms to investigate the morphological pathway of the origin of CPV-IIs [Fig 6A–D]). As previously mentioned, a CPV-II is defined as an alphavirus-induced vesicle with NCs bound to its membrane that predominate during mid- to late-stage infection in mammalian cells. In these tomograms, we observed 353 discrete vesicles that can be categorized as CPV-II. After segmentation and 3D visualization of each CPV-II, we identified four distinct CPV-II morphological forms. First, class 1 CPV-II were defined as unilamellar

vesicles with NCs bound to their cytoplasmic surface (Fig 6A and Video 6). Class 2 was composed of vesicles containing an invagination with NCs bound to the membrane in the invaginated space (Fig 6B and Video 7). Class 3 contained bilamellar vesicles, with NCs bound to the inner membrane and variably present on the outer membrane (Fig 6C and Video 8). Finally, class 4 vesicles were defined as single-membrane vesicles with enveloped virus particles inside and NC variably present on the outer membrane (Fig 6D and Video 9).

### Intracellular distribution of CPV-II

To investigate if the subcellular distribution of the four CPV-II classes follows a specific pattern, they were next analyzed for spatial distribution, size, and distance from the nearest Golgi stack. To increase the sampling size of each of these four classes and to understand their distribution, a second tomogram (Fig 6E b) was collected that overlapped with the first tomogram (Fig 6E a), extending up to the PM (Fig S4). 3D data on the CPV-II types were then collected from the joined segmented tomogram (Fig S4D). Each CPV-II was then color-coded by its morphological class and shown with the *trans*-most Golgi cisterna of the relevant Golgi stacks and the PM for spatial reference (see also Fig 6F and Video 10). Of the 353 CPV-II, class 1 was the most abundant (145 vesicles, 41%), followed by class 3 (90, 26%), class 4 (78, 22%), and class 2 (40, 11%) (pie chart in Fig 6F). The four classes of CPV-II were also analyzed for distance from the nearest Golgi stack to determine the occurrence of these classes in CPV-II clusters proximal to the Golgi and at the PM (Figs 6F and S5D).

The sizes of classes 1, 3, and 4 were also analyzed by calculating the volume of the segmented volumes for each vesicle using IMOD (Kremer et al, 1996) and converting volume measurements to diameter assuming a spherical shape (Fig 6G). For class 3, the outer membrane was used for size calculations. CPV-II in classes 1, 3, and 4 had similar diameter distributions for the interquartile range, with

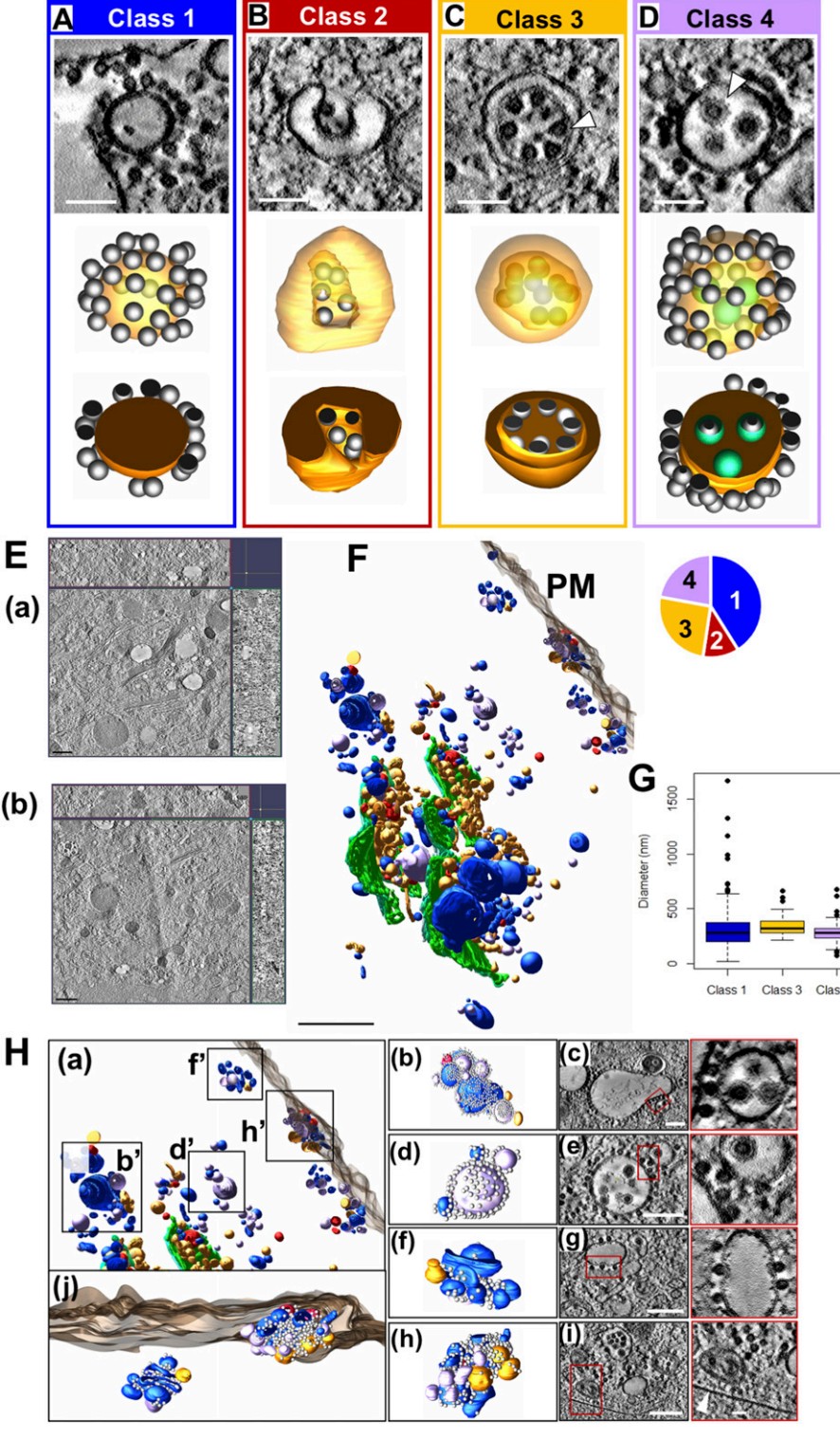

**Figure 6.  Classification and 3D analysis of four morphological forms of CPV-II.**
**(A, B, C, D)** A representative vesicle from each of the four CPV-II classes observed in the large-volume tomogram is shown in three views: 3-nm thick virtual sections (top image), full 3D models made partially transparent (middle image), and 3D cutaway models (bottom image, See Video 6–Video 9). NC (C, arrowhead) is distinguishable from fully enveloped virus particles (D, arrowhead). **(E)** A 3-nm thick virtual section from tomogram 1 and tomogram 2 ((a) and (b), respectively) with z-depth along the x- and y-axis shown along the top and right side, respectively. These were joined together for analysis of CPV-II distribution (See also Fig S7). **(F)** Spatial distribution of segmented CPV-II forms in the combined tomographic volume is shown in reference to the *trans*-most Golgi cisternae (green) and the plasma membrane (PM) (translucent gray). CPV-II are color-coded by class: class 1 (blue), class 2 (red), class 3 (gold), and class 4 (lavender) (see Video 10). The pie chart in the upper left shows the fraction of vesicles in each class. **(G)** For the three classes of roughly spherical CPV-II, the volume of each vesicle was calculated from the segmented mesh, and a diameter was approximated for each vesicle. Size distributions for class 1, 3, and 4 CPV-II are represented as a box plot with outliers representing data points beyond 1.5 times the interquartile range. Scale bars: 100 nm (A, B, C, D) and 1 μm (E). **(H)** CPV-II clusters in the cytoplasm and at the PM indicated by b′, d′, f′, and h′ in panel (a). Several vesicles in the cluster (h-i) come in proximity to the PM (white arrowhead (i)). Magnification of the region outlined in red (c), (e), (g), and (i) depict the interface between vesicles within the cluster. In (c) and (g), multiple NCs bind two separate CPV-II membranes through bivalent interactions; an observation that may explain the clustering of CPV-II. Scale bars: 200 nm, (c, e, g, i).

means of 326, 340, and 286 nm, respectively. Class 1 exhibited a notably broad size distribution with diameters ranging from 22 to 1,667 nm (Fig 6G), although a few extremely large vesicles were observed. Class 3 had no vesicles smaller than 214 nm and a narrow size range up to 662 nm.

CPV-II are distributed all over the cytoplasm including the inner leaflet of the PM. Interestingly, CPV-II exists primarily as clusters, especially at the PM. These clusters were shown previously to carry the Golgi marker α mannosidase-II ratifying their Golgi origin (Fig 1B and C). A careful analysis of the clusters

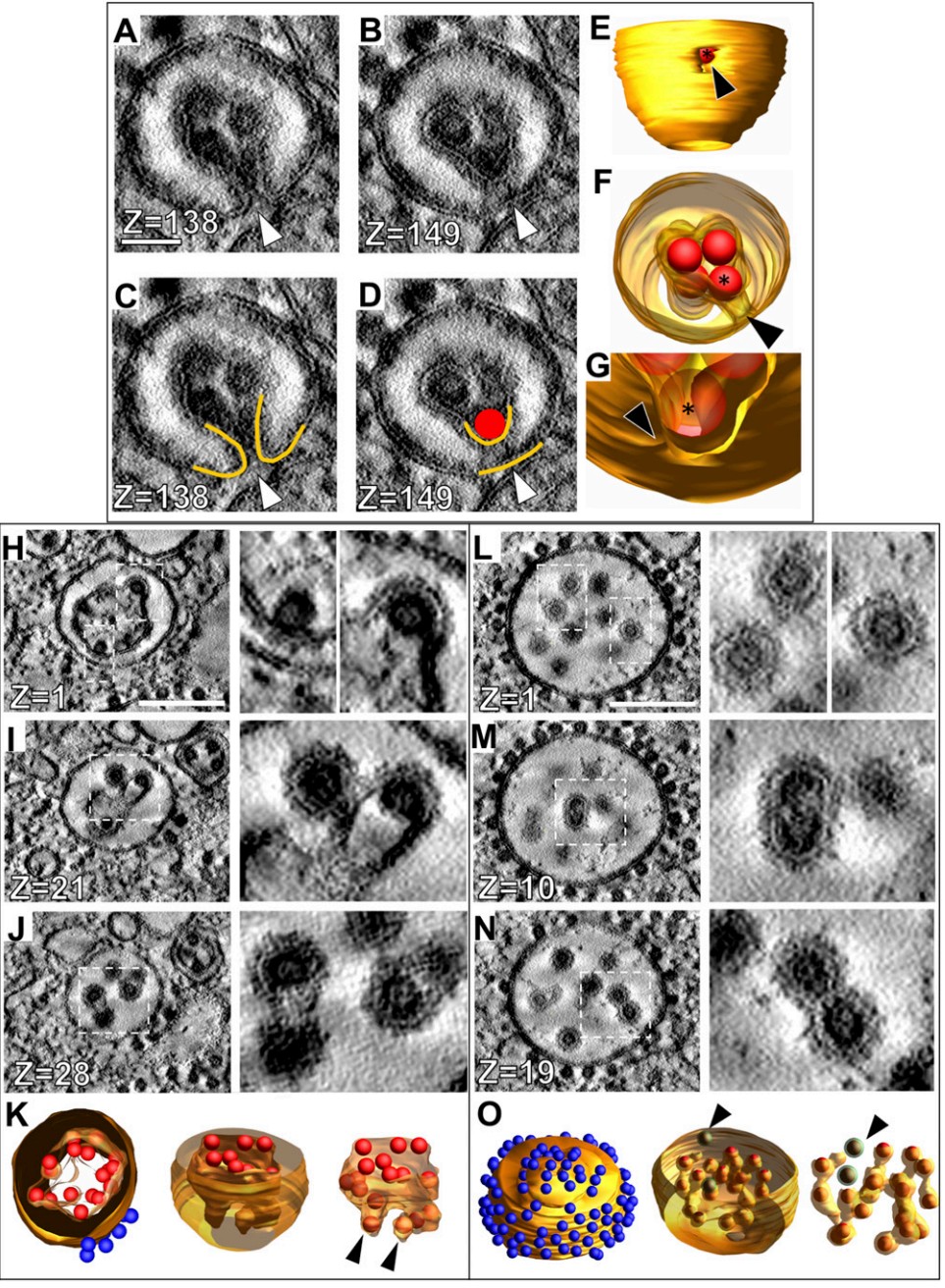

**Figure 7. 3D analysis of intermediate forms between classes 2, 3, and 4 reveals a potential maturation pathway.**
**(A, B)** Tomographic slices through an intermediate between class 2 and class 3 CPV-II. Each z-slice is 3 nm in thickness, with the slice number shown in the lower left corner of each image. **(A, C)** The fusion pore is indicated by the white arrowhead in (A) and (C). **(A, B, C, D)** Tracing of the vesicle membranes (gold) in (A, B) and a NC (red sphere) near the fusion pore. **(A, B, C, D, E, F, G)** 3D rendering of the vesicle shown in (A, B, C, D) with the fusion pore indicated by a black arrowhead. See Video 11. **(H, I, J, K)** Tomographic slices at various z-heights of a CPV-II intermediate between class 3 and class 4, which shows various stages of interior NC envelopment and moderate amounts of budding (H) to nearly complete envelopment (I, J). White dotted lines delineate areas magnified to the right. **(H, I, J, K)** 3D rendering of the vesicle shown in (H, I, J). See Video 12. **(L, M, N, O)** Tomographic slices through a class 4 CPV-II containing enveloped virus particles (L), dual-core particles (M), and enveloped NC connected to each other by membrane (N). **(L, M, N, O)** 3D rendering of the vesicle shown in (L, M, N), illustrating the interconnected nature of the partially enveloped NC. Completely enveloped NC without any membranous connection is indicated by the green segmentation and black arrowhead. See also Video 13. Scale bars: 50 nm (A) and 200 nm (H, L).

revealed that irrespective of its location, surface NC within these clusters made bivalent contacts between two adjoining CPV-II possibly holding the cluster together. Two representative clusters closer to the Golgi (Fig 6 (a)b', (b), (c), d', (d), and (e), respectively) and two by the PM (Fig 6H (f'), (f), (g), h', (h), and (i), respectively) are shown. CPV-II clusters at the PM (depicted in Fig 6H (a)) are shown as a segmented model from different point of view (Fig 6H (j), see Video 10).

## 3D analysis of intermediate CPV-II forms reveals a potential maturation pathway

In addition to the four classes of CPV-II just described, forms exhibiting intermediate features between the defined CPV-II classes were also observed. We therefore examined these forms closely for obtaining clues about their relatedness. First, the forms exhibiting characteristics of both class 2 and class 3 were examined.

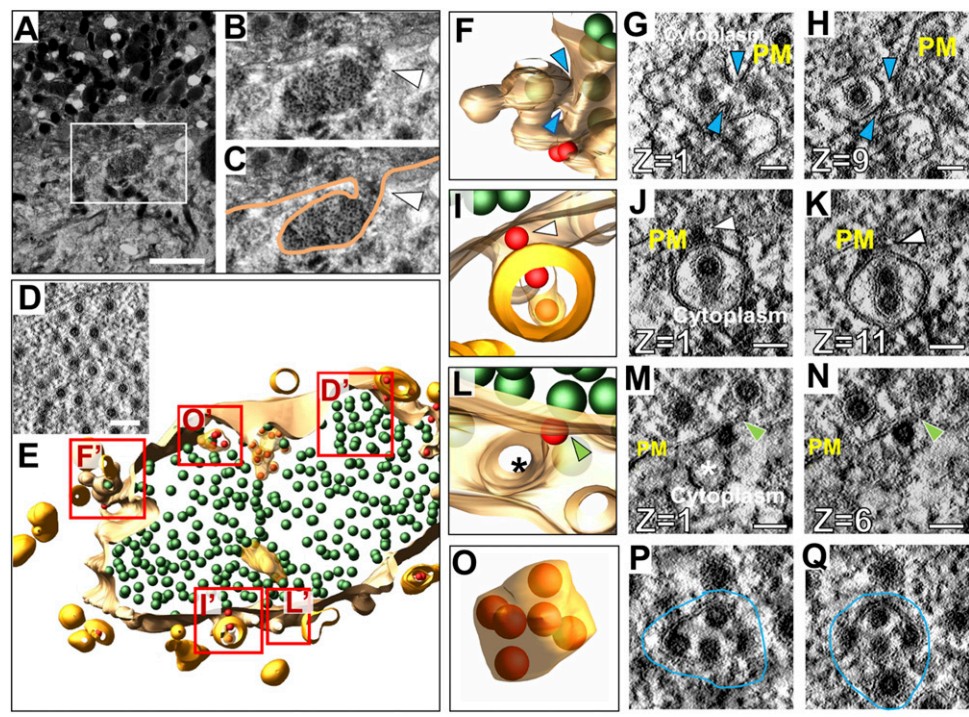

**Figure 8. Interaction of CPV-II with the inner leaflet of the plasma membrane (PM).**
**(A)** Transmission electron microscopy image of an area of a large vesicular structure connected with the PM via a narrow neck/tunnel like extension. **(A, B)** Magnified area of the yellow-dotted outline in (A). White arrowhead indicates the opening to the extracellular space. **(B, C)** Tracing of the PM shown in (B). **(A, B, C, D)** Tomographic slice through the sac-like structure shown in (A, B, C), revealing that it is filled with virus particles. **(E, F, G, H)** Segmented model of a 250-nm thick slice of the sac-like structure, with areas labeled D′–O′ magnified in panels (F, G, H). 3D rendering and slices through different z-heights of a class 4 CPV-II fusing to the PM. Blue arrowheads indicate the continuity between the vesicle membrane and the PM indicating a neck-like opening at different z-heights (Video 14). **(I, J, K)** Another class 4 CPV-II bound to the PM via a surface NC. White arrowheads indicate the NC simultaneously binding the PM and the CPV-II membrane, Video 15. **(L, M, N)** A class 1 CPV-II with a NC in the initial stages of budding through the PM. **(M, N)** Green arrowheads in (M) and (N) indicate the bending of the inner leaflet of the PM at the NC binding site. **(O, P, Q)** An extracellular vesicle containing incompletely budded NCs in various stages of envelopment (outlined in blue) found within the sac-like invagination. Scale bars: (A) 2 $\mu$m, (D) 100 nm, (G, H, I, J, K, L, M, N, O, P, Q) 50 nm.

A representative vesicle is depicted in Fig 7A–G (and Video 11) containing small areas of horseshoe-shaped cross-sections like class 2 (Fig 7A and C, compare with Fig 6B), whereas most of the vesicle cross-section resembled class 3 (Fig 7B and D, compare with Fig 6C). 3D visualization of the vesicle clarified the largely double-lamellar nature of the vesicle, apart from a small pore-like opening from the inner NC-containing compartment to the extravesicular space (Fig 7E arrowhead). The opening consisted of a narrow channel (Fig 7F and G arrowheads) with a single NC present inside the opening (asterisk in Fig 7E–G).

Apparent intermediates between class 3 and class 4 CPV-II carrying partial characteristics of both were also observed and analyzed in 3D. Tomographic slices through one such vesicle revealed slices resembling class 3 bilamellar vesicles (Fig 7H), intermediate slices with NCs which appear to be budding into the inner membrane of the vesicle (Fig 7I), and other slices with nearly enveloped virus particles resembling class 4 CPV-II (Fig 7J). 3D visualization revealed a single inner compartment with many of the NC in various stages of apparent envelopment, but all still connected to the main compartment (Fig 7K, arrowheads, Video 12). The wrapping/envelopment of NC by the inner membrane of class 3 CPV-IIs became very apparent in 3D, and this remodeling of the inner membrane around the NC trapped within was a common observation in the members of class 3. This observation suggested that the double-lamellar structure may not be the final end-product of this pathway and that further remodeling of the inner membrane possibly occurs.

A second intermediate between classes 3 and 4 is shown in Fig 7L–O (see also Video 13). In the 2D tomogram slices, this vesicle appears to be a class 4 vesicle with mature enveloped virions within. However, membrane connections between nearly all the membrane-wrapped NCs occur, indicating that they are not mature virions as concluded initially from thin-section screening (Fig 7M and N). Segmented tomograms clearly show the membranous connections between almost all the wrapped NCs except for two (Fig 7M and N, NC in green shown with an arrowhead in Fig 7O), which lack detectable membrane connection with the rest of the membrane wrapped virions.

To determine if CPV-II clusters play a role in virus egress, we screened the cell periphery to visualize CPV-II fusing with the PM. A large (~1 × 1.5 $\mu$m) sac-like structure was visualized at the PM filled with a high density of virions (Fig 8A–E). This sac-like structure revealed a continuity with the PM and an opening to extracellular space via a narrow ~700-nm long neck (Fig 8B and C, white arrowhead). Various forms of CPV-II were seen closely interacting with the inner membrane of this structure (Fig 8E). Careful scanning of the different Z slices revealed a class 4 CPV-II in what appeared to be a fused state with the PM (Fig 8F–H and F′ in E, see Video 14 and Video 15). A second class 4 CPV-II with a dual-cored virus particle was seen bound to the PM via an external NC (segmented in Fig 8I and its tomographic slices in Fig 8J and K). A class 1 CPV-II was also observed bound by an external NC to the PM (segmented in Fig 8L and in its tomographic slices in Fig 8M and N, respectively). The PM at the site of interaction with the NC exhibits bending, possibly indicating an initial stage of budding (Fig 8M and N, green arrowheads). Despite these observations, fusion-like events were rare (<10) in occurrence in tomogram 2. Segmentation revealed mostly fully formed free virions within the sac. In addition to virions, several single-lamellar vesicles containing multiple NCs in various stages of envelopment were also observed that were reminiscent of incompletely enveloped NCs within the lumen of the class 4 CPV-II (compare Fig 8O–Q with Fig 7L–O).

# Discussion

In this work, we sought to answer the question of how the stacked cisternal architecture of the GA is transformed into the vesicular forms of CPV-II during alphavirus egress. This structural remodeling of cellular secretory organelles infected with enveloped virus overlaps temporally and mechanistically to its egress pathway (Risco et al, 2003; Salanueva et al, 2003; Welsch et al, 2007, Welsch et al, 2009; Johnson & Baines, 2011). The viral components and interacting host proteins linked directly to its egress pathway has been the focus of intense research for decades; however, ultrastructural studies revealing the structural manipulation of the secretory organelles is relatively scarce. Such studies are critical for understanding the viral subversion of the host cell secretory system during egress and clues for structure–function relationships of these highly structured organelles.

To reach the aforementioned goal, the flexibility of repeated data collection from the same sample/cell is key. Thus, resin-based electron tomography (ET) of cryofixed sample was pursued and optimized (Ladinsky et al, 1994, 1999; Perkins et al, 1997; Miranda et al, 2015). A similar approach (ET of serial sections or ssET) was previously employed to image up to an entire Golgi ribbon at a resolution of 15–20 nm (Marsh et al, 2001; Noske et al, 2008). Here, we implemented an improvised phased data collection from 250-nm resin serial sections and used the montaging function of the serial EM software to cover a large area at a targeted magnification. Traditional EM imaging and Golgi-marker localization indicated the GA origin of the morphological forms of CPV-II. However, we were mindful that the GA, especially the trans-Golgi, is a major sorting junction where vesicles from the PM and the endosomes constantly interact. Thus, although the structural genesis of CPV-II may begin at the GA, it may contain some membrane components and even markers from membranes such as endosomes or the ER, albeit at low levels. This is especially true if the GA still retains its canonical functions within the context of VEEV infection and CPV-II biogenesis. As a result, we have kept this manuscript focused and limited on the contribution of the GA in the biogenesis of CPV-II. A more comprehensive 3D marker study is ongoing that aims to dissect the contribution of other organelles in the biogenesis of CPV-II.

Prior work in the field has employed traditional thin-section EM and was able to identify the unique NC-studded CPV-II structure (Mussgay & Weibel, 1962; Grimley et al, 1968; Soonsawad et al, 2010). However, the information obtained from these studies by the discrete and limited sampling of membrane could not establish the direct relationship of the cytopathic vesicles (CPV-II) with the GA. Thus, to detect these putative early changes and to determine the relationship of CPV-II with the GA, we carried out HRP-tagged marker studies employing a recently published method for significant sample preservation during localization of peroxidase-tagged proteins (Sengupta et al, 2019). Using HeLa cells constitutively expressing HRP-ManII, we demonstrated that this GA perturbation during alphavirus infection is neither cell type–specific nor an artefact resulting from overexpression of HRP-ManII. However, two things were apparent from our thin section 2D analysis; first, the GA begins to lose its cisternal architecture via widespread herniations and vacuolization leading to the formation of large

pleomorphic vesicles as early as 3 h PI. Second, thicker serial sections showed that CPV-II form large clusters that could have extensive connections with each other and may not all be isolated vesicles as previously thought (Mussgay & Weibel, 1962; Grimley et al, 1968).

We carefully analyzed multiple examples of each of these subclasses to look for intermediate forms that showed characteristics of two or more classes. The first tomogram covered a large subcellular volume consisting of multiple Golgi stacks where en bloc remodeling of cisterna was apparent. To record these changes systematically, our analysis progressed outward step by step from the intact *trans*-most cisternae. The predominant CPV-II observed in our thin-section (Fig S2) Golgi marker study (Fig 1) and in our tomograms (Fig 6) may originate from vacuolized Golgi cisterna as described previously by Griffiths et al (1983). The other type of Golgi remodeling distinct from the former results in CPV-II classes 2, 3, and 4. We provide 3D ultrastructural evidence that this second type originates from the bending and curling of intact Golgi cisternae. Transitional intermediates of various sizes exhibiting part cisternal and part bilamellar vesicular characteristics on the Golgi further substantiates this initial phase of the proposed morphological pathway (Figs 3Q–S and 4E–H).

Further away from the GA, independent CPV-II were analyzed in 3D for putative links. Intriguingly, many of these vesicles that at first appeared to be classes 2, 3, and 4 CPV-II were found to exhibit characteristics of two classes simultaneously. The ultrastructural evidence points to class 2 as the precursor of class 3 (Fig 7). These intermediates seem to have been captured at a stage just prior to end-to-end fusion giving rise to bona fide class 3 double-lamellar CPV-II. This characteristic of bent Golgi cisternae to form bilamellar structures by end-to-end fusion to create a compartment within a compartment has also been recently reported in brain tissue, where these Golgi structures function as degradosomes (Fernandez–Fernandez et al, 2017).

The morphological relationship between class 3 and class 4 may be the most intriguing of all. Fig 7H–K shows a typical example of such a transitional CPV-II that were in some of the 2D slices. This resembles class 3 CPV-II where the NC are just bound to the inner membrane without any visible distortion of the membrane or budding. However, advancing through the z-slices for the same CPV-II, it becomes apparent that the inner membrane at the point of contact with the NC exhibits bending consistent with various stages of alphavirus NC envelopment (Video 12 gives a great example of this feature). This observation led us to carefully examine the lumenal content of class 4 that apparently contains fully enveloped virions (Fig 7L–O). To our surprise, we observed that only two of the enveloped NCs within this CPV-II are independent virions. The rest of the seemingly matured virions were found to be connected to each other via intact membrane connections. This indicated that either their envelopment was incomplete at the time of cryofixation, or they never mature to form independent virions and that so far these membrane connections have gone undetected. Thus, our model proposes that classes 2, 3, and 4 CPV-II are all intermediates of the same morphological pathway that gives rise to enveloped/membrane-wrapped virions during alphavirus infection in mammalian cells (Fig 9A and B).

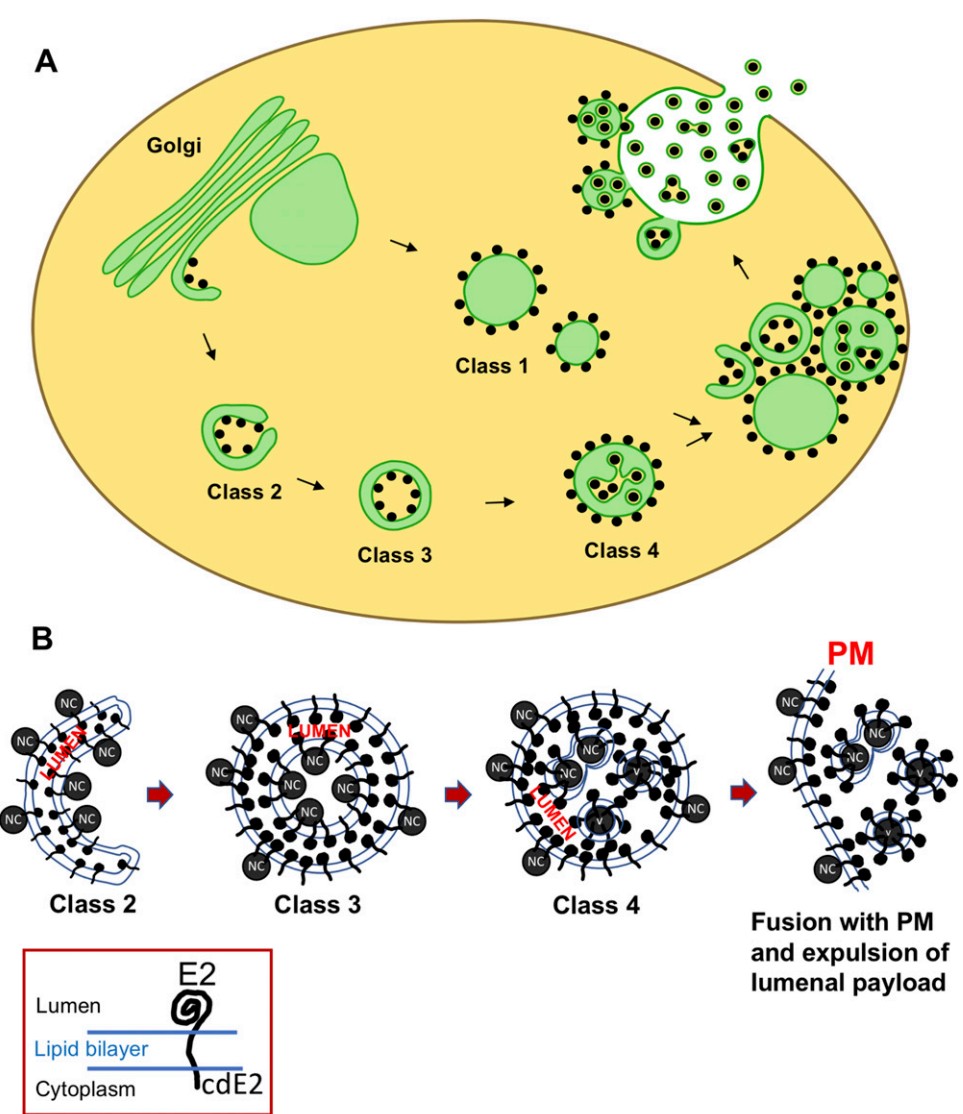

**Figure 9. Model of Golgi remodeling resulting in the biogenesis and maturation of CPV-II.**
**(A)** Proposed model illustrates that despite the distinct presence of four morphological classes of CPV-II; potentially two mechanisms operate. The morphogenesis of classes 2, 3, and 4 begins with the Golgi cisternae bending and fragmentation. Class 1 CPV-II, on the other hand, forms directly by herniation and swelling of the Golgi cisterna. Class 2 (or the sickle form) bends onto itself and fuses to give rise to the double-membrane vesicle structure (DMV) of Class 3. Finally, NC trapped within the DMV acquires its envelope from the inner membrane of the DMV to result in class 4. Finally, all these forms cluster via bivalent interaction of surface NC that bridges numerous CPV-II together and are trafficked to the plasma membrane (PM). From our data on events at the PM, we speculate that the NC on the surfaces of CPV-II is first transferred to the inner leaflet of the PM from where they bud out. In parallel, the CPV-IIs much smaller in size, fuses and delivers the lumenal virions to the cell exterior. **(B)** Hypothesizes the orientation of the E2 glycoprotein in the different classes of CPVII and the presence of NC and their potential delivery to the PM.

To lend more credibility to our hypothesis of inter-class conversion and maturation of CPV-II, we calculated the frequency of each class (class 1 in Fig S7A, class 2 in Fig S7B, class 3 in Fig S7C, and class 4 in Fig S7D) as a function of their distance from the Golgi (Fig S7E). It was observed that the largest class (class 1) remained the predominant class, and its counts comparable irrespective of its distance from the GA. However, as the distance from GA increases, the class 2 and 3 CPV-II counts fell considerably, whereas the count for class 4 increased. This observation supports our class conversion hypothesis in two ways. One, that classes 2 and 3 are possibly intermediate forms that mature to class 4 as they are trafficked toward the PM in clusters, and two, the morphogenesis of class 1 occurs by a separate mechanism and not by bending of the Golgi cisterna. Indeed, our results from the thin-section screening, Golgi marker localization, and our tomograms provide evidence that class 1 could in fact form from swelling, vacuolization, and fragmentation of the Golgi cisterna observed from a very early stage of infection (Figs 5A, B, and H, S2A (iii–iv), and S4A (i–vi) and B (v–vi)).

One apparent feature of the CPV-II analyzed within these overlapping volumes is that they exist as large clusters, more so at the PM (Fig 6H). CPV-II within these clusters is apparently tethered via bivalent interaction of NCs (Fig 6H (b–i)). Thus, a series of such interactions between multiple core-carrying vesicles could result in the formation of an elaborate vesicular cluster that can traffic as one. This mode of trafficking would be advantageous to the virus as it serves as an efficient mechanism for delivering large reservoirs of both viral glycoprotein and NCs from the GA to localized sites on the PM creating micro-domains rich in viral structural protein to facilitate PM budding. Recent evidence also demonstrated the requirement of actin remodeling proteins ARP3 and RAC1 in the later stages of infection as well as an actin–E2 interaction, indicating the CPV-II trafficking to the PM are actin dependent (Radoshitzky et al, 2016).

To gain a quantitative insight in this putative intracellular envelopment in alphaviruses, we investigated the potential efficiency of virus envelopment via this process. To this end, the surface areas of the inner compartment of class 3 CPV-II were measured, and the number of NCs inside was counted to determine if there is enough membrane for the internal NCs to become fully enveloped. We approximated that 5,000 nm$^2$ (dotted line, Fig S5A) of the membrane is required to envelope each NC (approximation based on a 40-nm diameter of NC). For class 3 CPV-II, we plotted the lumenal membrane surface area available per NC versus the percent of NCs that were undergoing potential budding (Fig S8). For each NC bound to the inner membrane, budding was determined based on the radius of curvature of the membrane at the point of attachment (Fig S6A–C). In general, the more membrane available, the more NCs exhibited budding; however, 53% of the CPV-II (9/17) did not have sufficient membrane to envelope all internal NCs (Fig S7). Thus, it is estimated that half of all NCs inside double-lamellar CPV-II could not become fully enveloped virus. This may point to the fact that this could be a dead-end byproduct of the formation of bilamellar CPV-II.

Several intriguing questions emerge from our results. From the virology standpoint, the primary question is: Does the pathogenic conversion of GA have bearing on the virus egress pathway? The answer obtained through this study is not clear cut. Our findings indicate that exocytic expulsion of internally budded virions is probably not a very efficient process judging by the number of class 4 CPV-II observed in the state of fusion (<10) with the PM. Interestingly a higher frequency of NCs on the outer surface of CPV-II (class 1 to class 4) are observed making direct contact with the PM and bending the inner leaflet of the PM (Fig 8I–N and represented in the model, Fig 9). One way to explain these observations is that the CPV-II can potentially mediate two pathways of viral egress at this stage of infection. First, the NC on its surface could potentially be transferred to the inner leaflet of the PM inducing PM budding, and second, this initial interaction of the CPV-II via the NC on the surface could in turn facilitate the fusion of the CPV-II to the PM and expulsion of the lumenal virus payload. In summary, despite the apparent inefficiency of the exocytic pathway, the CPV-II cluster that carries much greater number of NCs on its surface than in its lumen brings in a large amount of NCs and envelope glycoproteins to the PM that at least feeds into the major budding pathway. Lastly, it is also possible that the low number of budding/fusion events detected at the CPV-II clusters could be a result of the sample preparation. In this regard, cryoEM methods are being pursued to carefully screen the PM at the same stage of infection to rule out the possibilities.

As tools for visualizing virus induced membrane modifications in situ gets more and more sophisticated, more examples of viruses with multiple modes of budding and egress are gradually coming to light (Spiropoulou, 2001; Ghosh et al, 2020; Saraste & Prydz, 2021). In several virus families such as alphaviruses, whether a virus buds into intracellular membranes or at the PM is dictated largely by the interaction between the cytoplasmic domain of the viral glycoprotein (spike) and the cytoplasmic NC. For example, in alphaviruses, chemically induced accumulation of the glycoprotein at the Golgi results in the binding of NC directly to the Golgi and budding-like events (Griffiths et al, 1983). Thus, in our study, the late stage in

VEEV infection could represent a stage where natural accumulation of glycoprotein at the Golgi results in the binding of NC and formation of CPV-II. This in turn could be the mechanism by which the virus induces a more efficient co-trafficking of glycoproteins and NC to the PM in these late stages. In this regard, the CPV-II cluster may not be directly fusing to the PM and could rather be seen as a virus-induced "Golgi outpost" that could make the transfer of NC and glycoprotein more efficient as more and more viral proteins build up within the host cell. Further sorting and budding possibly operates within these clusters. Two observations support this theory; one, the absence of ManII-HRP signal on the PM in our DAB-based localization indicates the lack of/inefficient fusion of large CPV-II clusters that we show carrying the enzyme and two, the absence of direct visualization of multiple fused CPV-II with the PM despite the presence of multiple clusters. In addition, the CPV-II involved in the rather infrequent fusion events (<10) with the PM are around 100 nm in diameter, indicating that a further maturation and sorting state could be operating from these CPV-II clusters.

In this context, it will be important to investigate whether the CPV-II that are technically remodeled Golgi cisterna still retain GA secretory functions. Remodeling of the GA to create secretory Golgi outposts as result of changes in secretory demands of cells has been reported (Hanus & Ehlers, 2008; Kreft et al, 2010; Višnjar et al, 2017). Thus, mobilized Golgi cisternae in the form of CPV-II that clusters near the PM could be an example of viral subversion of this program to induce secretory outposts that function to meet the elevated level of structural protein transport to PM during budding. Either way, the study of GA remodeling in viral infection is also critical in understanding viral pathogenesis. Suppressive effects on the endogenous secretory system could lead to the inhibition of inflammatory cytokines, antiviral interferons, and cell surface expression of MHC-1. Remodeling and inhibition of the secretory pathway has been cited as a mechanism for host immune evasion by picornavirus (Dodd et al, 2001).

Mechanistically, the formation of curved Golgi cisternae leading up to its end-to-end fusion has one common denominator that cannot be ignored; all these structures have numerous NC bound to them. Furthermore, even a solitary NC bound to Golgi membranes exhibit membrane bending. Several mechanistic hypotheses could be forwarded to explain this phenomenon centering around the E2 glycoprotein, the NC, and their potential interaction. Work with bunyavirus glycoproteins has shown that viral glycoproteins by themselves can remodel and bend membranes and vacuolate the Golgi (Gahmberg et al, 1986). Thus, the phenomenon in alphavirus could be a result of accumulation of envelope glycoprotein, and the binding of NC is just coincidental. However, it is tempting to hypothesize that the interaction of capsid and E2 on the membrane leads to this cisternal curvature. It is also possible that the intrinsic architecture of the Golgi could be a contributor. Recent work using cryo-electron tomography has shown that the "flatness" of the Golgi cisternae is due to the presence of closely spaced yet unidentified lumenal proteins that zipper the membranes together into close apposition (18–19 nm) in the compact zone of the GA (Engel et al, 2015). The availability of the glycoproteins in the cisternal membrane leads to indiscriminate binding of the NC. Once the membrane associated E2 engages with the NC, it is possible that forces that otherwise induce budding into a herniated Golgi lumen

end up bending the zippered double-lamellar structure of the cisternae due to architectural constraints (Fig 9). To this end, we measured and compared the lumenal width of a typical cisternae and a double-lamellar CPV-II with or without detectable budding events (Fig S9A and B). The results indicate that "budding like" events were always associated with increased lumenal width (herniations), indicating that the Golgi architecture may play a role in this unique phenomenology in alphaviruses (Fig S9C–G). Alternatively, it could simply be that "wrapping" (Fig S9A and B) occurs when in situations where lipids can be replenished, whereas "budding like" phenomenon occurs when lipids are limiting in the double-membrane vesicle environment, leading to depletion of the inner membrane of the double-membrane vesicle to form a single-membrane vesicle.

In summary, we have combined multiple TEM approaches to elucidate host cell Golgi remodeling into various forms of CPV-II during VEEV infection. Using traditional thin-section TEM, we first established a timeline of structural changes of the GA during infection where the GA is converted into vesicular structures. We then applied an improvised hybrid sample fixation method in conjunction with peroxide-tagging of a Golgi marker to show that various forms of CPV-II in fact originate from the GA. To gain a 3D understanding of the process, we established a phased imaging approach for sequential imaging of overlapping volumes, employing serial-section electron tomography. We employed the automatic image montaging capability of the Titan-KRIOS operating at 300 kV at room temperature that enabled a larger volumetric coverage without compromising the target resolution (~4 nm). The vast array of 3D data collected enabled statistical analyses that yielded a 3D classification of the CPV-II system, identification of large and complex intermediates that links the various forms of CPV-II, relative spatial frequency of the various forms, and finally a model for the morphogenesis of CPV-II. Our data identify a second pathway of Golgi remodeling that accounts for the diverse morphological forms of CPV-II. Data from Golgi marker localization and reconstructed volumes also reveal that once the GA is converted into CPV-II, large clusters form that move out of the perinuclear region and accumulate at the PM where potential transfers of NC to the PM may occur. Thus, our work not only dissects a hallmark phenomenology associated with alphaviruses but also provides a path for utilizing similar resources to dissect remodeling pathways of large organelles associated in various diseases, endogenous cellular events, and infections.

# Materials and Methods

### Cell culture and virus infection

Baby hamster kidney cells (BHK-21, ATCC) were cultured in DMEM supplemented with 10% FBS at 37°C in a 5% $CO_2$ atmosphere. BHK cells were infected with the vaccine strain (TC-83) of VEEV at a MOI of 5, 10, 20, and 200. At various time points (3, 6, and 12 h), infected cells were released from Nunc UpCell dishes nonenzymatically. Cell sheets were pelleted, resuspended in 20% BSA (Sigma-Aldrich) as a cryoprotectant, and pelleted again for cryo-fixation and preparation for electron microscopy (described below).

### Localization of HRP-tagged Golgi marker, α-mannosidase-II

BHK cells were transfected with α-mannosidase-II-mCherry-HRP (Golgi FLIPPER, Kuipers et al, 2015) 8 h prior to infection. Infection was carried out as described above, followed by pelleting and fixation with 2% glutaraldehyde for 30 min, washing 3× for 5 min with 0.1% sodium cacodylate buffer and 1× with cacodylate buffer containing 1 mg/ml 3,3′-diaminobenzidine (DAB) (Sigma-Aldrich). Pellets were then incubated for 30 min in a freshly made solution of 1 mg/ml DAB and 5.88 mM hydrogen peroxide in cacodylate buffer, pelleted, and washed 3× for 5 min each in cacodylate buffer. Cell pellets were then resuspended in DMEM containing 15–20% BSA, pelleted again, and cryofixed by high-pressure freezing. Samples were prepared for electron microscopy as described below. For further experimental details on the CryoAPEX method, see Sengupta et al (2019). To determine if structural perturbation of Golgi induced by VEEV is cell type–specific and to rule out artefactual effects of overexpression of HRP-ManII, HeLa cells constitutively expressing the Golgi cis-medial marker α-mannosidase-II-HRP (kind gift from Dr. Franck Perez, Institut Curie) were infected with TC-83 at an MOI of 10 and 20, and cells we processed for EM identically as in the case of infected BHK cells transiently expressing the HRP-ManII plasmid.

### Sample preparation for electron microscopy and tomography

Cell pellets in cryoprotectant solution were loaded onto membrane carriers (Mager Scientific) and cryo-fixed using the EM PACT2 high-pressure freezer (Leica). Cryo-fixed cells were processed by freeze substitution using an AFS2 automated freeze substitution unit (Leica). Briefly, frozen pellets were incubated at –90°C in a solution of tannic acid in acetone, followed by slowly warming to room temperature in a solution of uranyl acetate and osmium tetroxide in acetone. Samples were infiltrated with Durcupan ACM resin (Sigma-Aldrich), and blocks were polymerized at 60°C. Thin sections (60–90 nm) were cut using the UC7 ultramicrotome (Leica), post-stained with 2% aqueous uranyl acetate and Sato's lead, and imaged on a Tecnai T12 microscope (FEI) operating at 80 kV. Thicker 250-nm sections were screened on a 200 kV (CM200 Philips) microscope. Samples from all time points were screened in both thin and thick sections. For tomography, serial sections of 250-nm thickness were collected on LUXFilm-coated 2 × 1 mm copper slot grids (Luxel), post-stained, carbon-coated, and overlaid with 10-nm colloidal gold particles (Sigma-Aldrich) for use as fiducial markers. At least 50 sections were screened for morphological determination from two representative resin blocks for each time point and condition.

### Large-volume electron tomography data collection

Two large-volume EM tomography data sets were collected covering an ~45 $\mu m^2$ area of a BHK cell through a depth of 2–2.5 $\mu m$ (Fig S2). Images were acquired using a Titan Krios TEM (FEI), operating at 300 kV at room temperature, outfitted with a 4k × 4k UltraScan 4000 CCD camera (Gatan, Inc.). Tilt series were collected with automation using the program SerialEM (Mastronarde, 2003). First, a low magnification image montage of the entire grid was created to use as a map for marking regions of interest. The area of interest in the cell was visualized and marked on consecutive serial sections.

At each section, a tilt series was collected. For each tilt series, images were collected at tilts from +60 to –60 degrees in 2° increments. To increase the area covered by each image, 2 × 2 montage images were collected at each tilt at 14,000× magnification, corresponding to a 0.65 nm/pixel size at the specimen level. Such montage tilt series were collected through 10 serial sections (Tomogram 1) or 7 serial sections (Tomogram 2). Smaller tomograms were collected at a pixel size of 0.8 nm, using a similar collection scheme, without serial sectioning or montaging.

### Electron tomography data reconstruction and analysis

Tilt series were aligned, and tomograms generated by weighted back projection using the eTomo interface of IMOD (Kremer et al, 1996). Serial section tomograms were aligned and joined in z to produce two reconstructed volumes with approximate dimensions 5 × 5 × 2.5 $\mu m$ (tomogram 1) and 5 × 5 × 2 $\mu m$ (tomogram 2). Membrane structures of interest were segmented by hand tracing using IMOD's Drawing Tools and Interpolator, and 3D surface mesh models were generated (O'Toole et al, 2018). NCs were segmented by placing a sphere with a diameter of 40 nm at the center of each density. Models for the two large tomograms were joined using cellular landmarks to determine translational and rotational parameters for alignment.

Size measurements for CPV-II were obtained from the 3D mesh of the segmentation for each vesicle. The volume inside the mesh for each CPV-II was calculated in IMOD, and volume was converted to diameter assuming spherical vesicles, for ease of comparison to existing CPV-II measurements. Nearest neighbor distance analysis between each CPV-II object and each Golgi object was conducted using the *mtk* program in IMOD. Classification of NCs was achieved by analysis of the radius of curvature on adjacent membranes of Golgi cisternae and Golgi-derived vesicles. The *imodcurvature* command in IMOD was used to identify areas on the segmented model with local radius of curvature between 20 and 60 nm, corresponding to the curvature produced by a budding 40 nm NC. These areas were colored red on the membrane, as shown in Fig S6. NCs adjacent to these highly curved membranes were colored red, whereas NCs adjacent membranes having a radius of curvature greater than 60 nm (represented by no change in color of the membrane) were colored blue.

Details and illustration of lumenal distance measurements in Fig S9C–G can be found in the figure legend. At least 60 measurements were taken for each group, and the Mann–Whitney U test was used to compare each pair of distributions. Statistical analysis was done using R software (R Core Team, 2019).

## Data Availability

The manuscript does not have large-scale data sets to deposit to public databases.

## Supplementary Information

## Acknowledgements

These studies were supported by the National Institutes of Health (R00RR026211 to JK Lanman, AI095366 to RJ Kuhn, and AI081077 to RV Stahelin). We thank Dr. Carolyn Machamer and Dr. Peter Hollenbeck and for their useful discussions and Dr. Saif Hassan for brainstorming sessions and help with figures. We thank Dr. Agustin Avila-Sakar and Valerie Bowman at the Purdue Cryo-EM facility for providing ready access to the Titan-KRIOS and CM200 microscopes and other instruments in the facility. We thank Dr. Christopher Gilpin at the Purdue Life Sciences Microscopy Facility for access to the T12 electron microscope. Lastly, we thank Anwesha Dasgupta for her help in editing the manuscript.

## Author Contributions

R Sengupta: conceptualization, formal analysis, investigation, visualization, methodology, and writing—original draft, review, and editing.
EM Mihelc: formal analysis, investigation, visualization, methodology, and writing—original draft, review, and editing.
S Angel: investigation and writing—review and editing.
JK Lanman: conceptualization, resources, formal analysis, supervision, funding acquisition, investigation, visualization, methodology, project administration, and writing—original draft, review, and editing.
RJ Kuhn: resources, supervision, funding acquisition, project administration, and writing—review and editing.
RV Stahelin: resources, supervision, project administration, and writing—review and editing.

## Conflict of Interest Statement

The authors declare that they have no conflict of interest.

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
