## [Reviewer comments · Life Science Alliance]

Life Science Alliance

Contribution of the Golgi apparatus in morphogenesis of a virus induced cytopathic vacuolar system

Ranjan Sengupta, Elaine Mihelc, Stephanie Angel, Jason Lanman, Richard Kuhn, and Robert Stahelin

DOI: <https://doi.org/10.26508/lsa.202000887>

Corresponding author(s): Robert Stahelin, Purdue University West Lafayette and Ranjan Sengupta

Review Timeline:	Submission Date:	2020-08-20
	Editorial Decision:	2020-09-25
	Revision Received:	2022-08-19
	Editorial Decision:	2022-09-06
	Revision Received:	2022-09-07
	Accepted:	2022-09-08

Transaction Report:

September 25, 2020

Re: Life Science Alliance manuscript #LSA-2020-00887-T

Prof. Robert V Stahelin
Purdue University West Lafayette
Medicinal Chemistry and Molecular Pharmacology
207 S. Martin Jischke Drive
Room 446 DLR Building
West Lafayette, IN 47907

Dear Dr. Stahelin,

Thank you for submitting your manuscript entitled "Morphogenesis of a virus induced vacuolar system from the host Golgi apparatus" to Life Science Alliance. The manuscript was assessed by expert reviewers, whose comments are appended to this letter.

As you will note from the comments below, the reviewers are generally enthusiastic about the manuscript and the findings, but have raised some concerns that should be addressed in the revised manuscript. Particularly, we request you to re-write the manuscript for focus (R2 and R3) and clarify that the results can not be regarded as a general mechanism for alphavirus infection and trafficking (R3). All other concerns raised by the referees should be addressed as well.

To upload the revised version of your manuscript, please log in to your account: <https://lsa.msubmit.net/cgi-bin/main.plex>. You will be guided to complete the submission of your revised manuscript and to fill in all necessary information. Please get in touch in case you do not know or remember your login name.

We would be happy to discuss the individual revision points further with you should this be helpful. While you are revising your manuscript, please also attend to the below editorial points to help expedite the publication of your manuscript. Please direct any editorial questions to the journal office.

The typical timeframe for revisions is three months. Please note that papers are generally considered through only one revision cycle, so strong support from the referees on the revised version is needed for acceptance. When submitting the revision, please include a letter addressing the reviewers' comments point by point.

Thank you for considering Life Science Alliance as an appropriate venue for your research. We look forward to receiving your revised manuscript.

Sincerely,

Shachi Bhatt, Ph.D.
Executive Editor
Life Science Alliance

B. MANUSCRIPT ORGANIZATION AND FORMATTING:

Reviewer #1 (Comments to the Authors (Required)):

This manuscript from Stahelin's lab is a well-done EM study, which provides new insight into the role of the Golgi in alphavirus infection and trafficking. The Authors use a wide repertoire of EM approaches (including fast-freezing, HRP labelling and EM tomography) to reconstruct the pathway underlying the biogenesis of a cytopathic vacuolar system (CPV-II) in alphavirus-infected cells. The EM analysis suggests that CPV-IIs form via bending of Golgi cisternae into double-membrane vesicles containing NCs, which mature into enveloped viral particles after remodeling of internal membrane. This study proposes a new model of CPV-II morphogenesis from the GA and could be of great interest to the readership of the Life Science Alliance. However, in my view, in its current form the manuscript lacks few bits and could be improved by addressing my comments below.

Major comments.

- 1) It is not clear whether CPV-IIs fuse with the plasma membrane of infected cells to release viral particles. At least class 4 CPV-IIs contain enveloped particles and might be involved in their release. Do authors observe any continuity between CPV-IIs and PM in tomograms? If not, probably later post infection time points have to be analyzed.
- 2) Along the same line, all CPV-IIs apparently contain Golgi marker (ManII). Are they mature enough to fuse with the plasma membrane for release of viral particles? Usually bona fide transport intermediates that carry proteins from the Golgi to plasma membrane do not contain resident Golgi enzymes (like ManII). Should ManII be recycled back to the Golgi from CPV-II to complete CPV-IIs maturation and make them PM-fusion capable? Again, analysis of the late post infection time points might help to answer these questions.
- 3) How CPV-IIs scatter through the cytoplasm? Do they use microtubules or other cytoskeleton elements?
- 4) The authors provide very convincing images showing different steps of CPV-IIs maturation through classes 2-4. This process leads to the formation of enveloped viral particles inside the lumen of class 4 CPV-IIs. In contrast, class 1 CPV-IIs seem to be useless for assembly of enveloped particles and their further release. Any explanation regarding how class 1 CPV-IIs might be implicated in the virus replication/release?
- 5) Class 3 CPV-IIs are double membrane vesicles (DMVs) and resemble autophagosomes. In this context, did the Authors try to test whether CPV-IIs contain autophagosome markers? Being DMVs autophagosomes and CPV-IIs might share some molecular machineries/components that drive their morphogenesis.

Minor comments.

- 1) It would be better to include images of the control Golgi in the main figure to show the difference between the organelle ultrastructure in control and infected cells.
- 2) Discussion. The authors say that FIB-SEM does not offer sufficient resolution for investigation of lipid bilayer remodeling in membrane organelles. This is not exactly the case because high-end FIB-SEM systems from Zeiss and Tecnai (Thermo Fisher now) resolve lipid bilayer in Golgi membranes, while 3D capabilities of these systems provide significant advantage over tomogram stitching. So, I would probably moderate this statement or remove it from the discussion.

Reviewer #2 (Comments to the Authors (Required)):

Sengupta et al use a range of high-powered electron microscopy techniques including 3D electron tomography to study the morphology of the Golgi apparatus after alphavirus infection. The combination of Golgi marker-peroxidase/DAB stained Golgi (Golgi FLIPPER) and high-pressure frozen serial section electron tomography are the most suitable and among the highest quality/resolution methods currently available to study large volume Golgi morphology in detail, and the quality of the images

and the analyses presented here make a significant contribution to our understanding of the pathway in at least this system. This study specifically provides novel insight into alphavirus infection in a number of areas that include;

- (i) Reshaping of the Golgi through cisternal alterations including their bending into double-lamellar vesicles,
- (ii) The complexity and interconnections within the pleomorphic complex cytopathic vacuolar system (CPV-II).
- (iii) Characterisation of different classes of CPV-IIs and the transition between some of these classes.
- (iv) Mobilization of Golgi membrane towards the plasma membrane.

Overall, this is a detailed, well executed and important study that sheds light on alphavirus infection and the impact of viral particles on reshaping the Golgi apparatus.

Main comments

The LSA instructions to authors seem to allow space for a proper discussion, and this paper is a bit light in this aspect- the data raise a number of questions that I suggest could be discussed by the authors. However, the authors have restrained themselves from speculation, reducing- in our view- the impact of their observations. They have refrained from discussing their work in the context not only of how these alterations might relate to the remodelling involved in replication, but also how CPV-II might transition to exocytosis, and finally of whether they believe that this data might apply to all viruses that bud into the secretory pathway to escape from the cell, obviously including how they would view this in relation to Sars-Cov-2.

The CPV-II are implied- including in the title - as originating from the Golgi only. In the discussion they conclude that "prevalent forms of CPV-II carried the Golgi HRP-marker ". These data only show that at least some of the membrane is from the Golgi, since the identity of a continuous membrane that is labelled by a protein in the bilayer may be a patchwork of contributions. The contribution from pre or post-Golgi organelles is not investigated here, nor discussed, despite two possible contributors; firstly the complex set of double-membrane structures thought to form from the ER and associated with replication, secondly the commonly seen contribution from endosomal membrane/proteins to secretory organelles - and the CPV is proposed to lead to viral release. Investigating how the ER or endosomal -derived membranes relate to the Golgi-derived structures seen here is not needed for this paper, but the relation of these structures needs at least to be discussed.

The topology of the structures is complex, and the location and orientation of the spike proteins and how these viruses are ultimately released by fusion of such complex structures with the plasma membrane is not addressed in the Model of Fig. 10, (where it would help the reader to keep track of how this critical aspect of viral formation relates to the membrane remodelling) nor in the text- perhaps not surprising since Fig. 10 is not mentioned in the text.

In Fig. 10, many of the NCs seem likely to be trapped in a dead-end structure. Indeed, one interpretation of Fig. 10 would be that only 2 virions in that diagram are going to be secreted free of wrapping membrane and ready to re-infect. This would be an extraordinarily wasteful process! Some speculation regarding membrane remodelling in late infection would be appropriate, or at least a clear statement that we are only seeing an intermediate stage and that there is another whole set of remodelling(s) required before release- if that is the authors view.

The language used in this paper is rather dry and difficult to follow, and readers would be helped by a thorough rewrite.

Further comments:

Evidence for accumulation of CPVII at the plasma membrane could be stronger. In the example shown in Figure 1 E(i) it is difficult to make out the plasma membrane. A dotted line or arrows would be useful to denote the position of the plasma membrane; this should be added to Figure 2 as well. To corroborate the example shown in Figure 2 B (i and vi) further examples should be included in supplementary.

Line 159- Mentions CPV-II studded with NC in Figure 1D ii, but these are very difficult to make out. Similarly, lines 164-165 mentions that "NC-studded surface of these structures is evident", which does not seem to be the case. Either modify these statements or the figure, replacing or adding zoomed panels where this is clearer.

Figure 5 A and B - Particularly in panel A; using two shades of green makes it difficult to differentiate between the two structures. An alternative to the dark green should be used or the authors could consider adding further images showing the green and dark green structures alone so that the reader can see the position and differentiate between the two.

There is a reference in the text to Fig 7H (line 389). This is not present in the figure that I have in the merged file that I am working with, and lines 391/392 also mention 7f' and other similarly named figures that are not present. Further, 393 has a sentence starting "First...." and never gets to second and maybe would have been all the way to tenth! lines 388 to 395 refer to panels in Figure 7 that do not exist, and it may be that this paragraph should be referring to Figure 8.

Line 89 - Introduce TEM acronym - transmission electron microscopy (TEM)

Line 131 - Should this be the white arrow rather than yellow for the stacked architecture?

Line 160 - "These clusters are seen 159 enmeshed and at times cradled by surrounding cisternae (Figure 1C ii and Figure 1D

iii)" is this shown by the blue arrows?

Line 166 - Should v be vi?

Line 209 - Should "Figure 2B ii and iv" be Figure 2B iii and iv?

Line 219 to 220 - When mentioning the mix of various morphological forms, consider referring to figure 7 where you have further characterised these and that class III is not shown in the Figure C iii.

Figure 3B - Could include XY, XZ and YZ labels in the panel for the different views of the tomogram. In the XZ view the Golgi that can be seen, is it part of stack 1 or 3? If so, include the label in the XZ window.

Figure 4 B and F - Dotted box width does not match the regions shown in C, D, G and H. G and H shows a larger area and lower magnification than C and D, hence the small NC sphere size and therefore it would be beneficial to the reader to have scalebars in B-D and f-H.

Line 377 - Figure 4F should be Figure 7F.

Should include a supplementary video of Golgi stack 1 and Golgi stack 2 showing slices from the tomogram transitioning into the model shown in Figure 4 A-B and E-F.

Sometimes refer to Fig and other times Figure - see line 139, 198, 203, 278, 288, 331, 335, 629.

Figure S6 is not referred to in the text.

Reviewer #3 (Comments to the Authors (Required)):

Manuscript provides detailed description how host cell Golgi complex undergoes conversion into pleomorphic cytopathic vacuolar system (CPV-II) during Venezuelan equine encephalitis virus (VEEV) infection using high resolution electron tomography. Authors use the state-of-the-art technology for morphological analysis, have used appropriate sample preparation methods to best preserve the organelle integrity, and provide extremely detailed analysis of the images.

Alphaviruses have similar size and genome, but vary in their pathogenicity, preferred host receptors and cellular sites for virus genome transcription, packing of virions and budding sites. Therefore, these results cannot be regarded as a general mechanism for alphavirus infection and trafficking. Although I admired the quality of the tomograms, conclusions reached too far as they were based only to a portion of pleomorphic CPV-II from one cell. Authors present a four-step pathway for CPV-II biogenesis, which is interesting, but too speculative. Moreover, manuscript was very heavy to read, as it lacked focus, contained too much details of work that could be considered as preparatory work, and was full of small writing errors. I consider this manuscript too immature and it's scientific impact too low to be suitable for publication.

Here are minor comments on how to improve the manuscript.

The manuscript could be made shorter and more focused throughout, for example by removing the comparisons to autophagosome formation in the introduction, results, and discussion, as there is no evidence to show that similar mechanism is used. In fact, there is lot of evidence of the opposite. Especially text in lines 407-417 is pure speculation, and should not be in the results section.

The beginning of the results contains too much results (e.g. screening of optimal MOI for virus infection and TEM screen from different time points post infection) that could be considered as preparatory work that need not to be reported with such details. Figure 1 is not informative; especially I cannot see any point for Figure 1C, unless the aim is to show that these sections exist. Figure 1A shows Golgi cisternae without any contrast, in comparison to most of the other images. Is this a processing artefact? Moreover, if the structures are devoid of NCs (lanes 133-134), how one can be sure that this is from an infected cell? NCs should be visible in 90 nm thin sections, as seen in other images too. Cisternal bending depicted in purple in Figure 1Biii is quite typical for Golgi cisternae. Figure 1D shows collage of projection images from 250 nm thick section with very low informational value due to overlapping structures. I would seriously consider leaving the whole Figure 1 out.

Figure 2 has very little membrane contrast, and trans cisternae cannot be seen at all. Is this in all cells? I have seen nicer morphology with standard embedding techniques.

lane 131: ...typical stacked architecture.. is depicted with white arrow (not yellow as in the text)

lane 192: TC-83 is mentioned here, and explained only in the discussion part

lane 198: instead of osmium stain, should it be DAB precipitate?

lanes 293-295: This part is very speculative, if there was there only one intermediate structure of this type

lane 389: I could not find Figure 7H

lane 393: ... two common features... What is the second one?

lanes 391-393: not clear which images are referred to

lane 402-402: comparison to Figures 4B and 4C does not make any sense. Should it be Figures 7B and 7C?

lane 436: NC in green shown with an arrow is in Figure 9O

lane 503: lipid bilayer thickness should be 4 nm and not microns

Abbreviation to post-infection (PI) is introduced on line 120, and then not used over 20 times after that. Similar inconsistencies with the use of many other abbreviations too.

We would like to thank the reviewers for taking the time to read our manuscript and provide critical and timely feedback on its shortcomings. We have revised the manuscript accordingly and have collected new data per the reviewers' comments. A point-by-point response to each reviewer query can be found below.

Reviewer #1 (Comments to the Authors (Required)):

This manuscript from Stahelin's lab is a well-done EM study, which provides new insight into the role of the Golgi in alphavirus infection and trafficking. The Authors use a wide repertoire of EM approaches (including fast-freezing, HRP labelling and EM tomography) to reconstruct the pathway underlying the biogenesis of a cytopathic vacuolar system (CPV-II) in alphavirus-infected cells. The EM analysis suggests that CPV-IIs form via bending of Golgi cisternae into double-membrane vesicles containing NCs, which mature into enveloped viral particles after remodeling of internal membrane. This study proposes a new model of CPV-II morphogenesis from the GA and could be of great interest to the readership of the Life Science Alliance. However, in my view, in its current form the manuscript lacks few bits and could be improved by addressing my comments below.

Major comments.

1) It is not clear whether CPV-IIs fuse with the plasma membrane of infected cells to release viral particles. At least class 4 CPV-IIs contain enveloped particles and might be involved in their release. Do authors observe any continuity between CPV-IIs and PM in tomograms? If not, probably later post infection time points have to be analyzed.

Response: We thank the reviewer for this inquiry and suggestion. We have added EM tomography data that shows continuity of CPV-II with the plasma membrane. However, these observations were very limited in the cells screened. We could suggest two theories to explain that result, first the CPV-II are perhaps not very efficient in fusing with the PM or may not be the primary vesicles that fuse with the plasma membrane but instead represents Golgi outposts at the plasma membrane from where smaller vesicles carrying glycoproteins are transported very quickly (to the PM). These vesicles are perhaps smaller than what could be detected by our method. The relatively smaller size of the CPV-II vesicles that shows continuity with the PM supports that theory. As a result, we are pursuing cryo-ET and cryo-CLEM to follow up on that hypothesis. Since our primary goal in this study was to investigate the Golgi apparatus for visualizing structural changes during the birth of CPV-II, the methods used were not aimed at getting the best view of the plasma membrane. For example, since cells were not cryofixed as a monolayer on a sapphire support it is possible that the edge of the cells could not be optimally imaged. We have screened samples at various time points before and after as suggested by the reviewer, but we could not see many CPV-II showing continuity with the inner leaflet of the PM.

2) Along the same line, all CPV-IIs apparently contain Golgi marker (ManII). Are they mature enough to fuse with the plasma membrane for release of viral particles? Usually, bona fide transport intermediates that carry proteins from the Golgi to plasma membrane do not contain resident Golgi enzymes (like ManII). Should ManII be recycled back to the Golgi from CPV-II to complete CPV-IIs maturation and make them PM-fusion capable? Again, analysis of the late post infection time points might help to answer these questions.

Response: We have screened infected cells expressing the HRP-tagged Golgi marker (ManII) at different timepoints but unfortunately, we could not detect a signal strong enough at the PM to make a claim with certainty. The best and the most unambiguous way to answer such a question is to use cryo-CLEM that will reveal both the nature of carriers, its contents, as well as presence of the marker protein on the PM. Unfortunately, it's currently beyond the scope of this manuscript. We are glad that the

reviewer raises this point that has struck us as very engaging and an important area of future research based on our current findings.

3) How CPV-IIs scatter through the cytoplasm? Do they use microtubules or other cytoskeleton elements?

Response: We have added this pertinent point in our discussion in the revised version. Recent evidence from work by Radoshitzky et al. demonstrate that actin remodeling proteins ARP3 and RAC1 are required in the late stages of VEEV infection, and that actin interact with the glycoprotein E2 directly. Ultrastructural evidence shows CPV-II to be in interaction with actin bundles within the cell, we see similar things time and again in our EM images as well. More work clearly needs to be done to understand the dynamics of this transport.

4) The authors provide very convincing images showing different steps of CPV-IIs maturation through classes 2-4. This process leads to the formation of enveloped viral particles inside the lumen of class 4 CPV-IIs. In contrast, class 1 CPV-IIs seem to be useless for assembly of enveloped particles and their further release. Any explanation regarding how class 1 CPV-IIs might be implicated in the virus replication/release?

Response: We thank the reviewer for this important question. Our ultrastructural snapshots and marker studies supports the hypothesis that class I CPV-II are in fact swollen Golgi cisterna. It is true that they do not fit into the scheme of things as far as intracellular budding of the NC is concerned. However, ultrastructural data that we have now added in the revised manuscript show the NC on the surface making direct contact with the inner leaflet of the PM and producing membrane bending. We therefore hypothesize that cores on the surface of CPV-II are deposited onto the inner leaflet of the PM from where they bud out. Incidentally, our calculations show that the class 1 CPV-II carries the highest number of NC. Over the years, endosomal origin of CPV-II have been speculated but we were unable to detect that in our preliminary marker screen. A more descriptive and in-depth work using membrane markers for other intracellular organelles is in the works. As discussed earlier, the primary goal of this work was to investigate the contribution of the host Golgi in the birth of CPV-II.

5) Class 3 CPV-IIs are double membrane vesicles (DMVs) and resemble autophagosomes. In this context, did the Authors try to test whether CPV-IIs contain autophagosome markers? Being DMVs autophagosomes and CPV-IIs might share some molecular machineries/components that drive their morphogenesis.

Response: While this is admittedly a great question for further dissection of the system unfortunately its outside our scope and the time limit imposed on us. Our early attempt to used autophagosomal marker (LC3-APEX2) did not give us a straightforward result with promiscuous staining of sites that have not been reported before. Thus, further work needs to be done to characterize such markers and needs to be compared with other autophagosomal markers in parallel and then applied on this infection system in question. For now, we hope we have provided comprehensive evidence for Golgi involvement in the biogenesis of CPV-II and the steps involved in its structural dismantling of a premier secretory organelle of the cell. Reviewer 3 also thought the autophagosome part we originally presented was too speculative. Thus, it has been removed from the revised text.

Minor comments.

1) It would be better to include images of the control Golgi in the main figure to show the difference between the organelle ultrastructure in control and infected cells.

Response: The images of the control Golgi has now been added to the main figure (Fig1A) as suggested by the reviewer. Authors thanks the reviewer for her/his suggestion.

2) Discussion. The authors say that FIB-SEM does not offer sufficient resolution for investigation of lipid bilayer remodeling in membrane organelles. This is not exactly the case because high-end FIB-SEM systems from Zeiss and Tecnai (Thermo Fisher now) resolve lipid bilayer in Golgi membranes, while 3D capabilities of these systems provide significant advantage over tomogram stitching. So, I would probably moderate this statement or remove it from the discussion.

Response: We appreciate the concern of the reviewer and recognize our previous statement wasn't accurate. We have taken that statement out of the revised version entirely.

Reviewer #2 (Comments to the Authors (Required)):

Main comments

1) The LSA instructions to authors seem to allow space for a proper discussion, and this paper is a bit light in this aspect- the data raise a number of questions that I suggest could be discussed by the authors. However, the authors have restrained themselves from speculation, reducing- in our view- the impact of their observations. They have refrained from discussing their work in the context not only of how these alterations might relate to the remodeling involved in replication, but also how CPV-II might transition to exocytosis, and finally of whether they believe that this data might apply to all viruses that bud into the secretory pathway to escape from the cell, obviously including how they would view this in relation to Sars-CoV-2.

Response: We appreciate the suggestion by the reviewer for a more elaborate discussion on the role of CPV-II in a secondary pathway of egress in alphaviruses and the egress of other viruses in light of the findings here. As a result, we have added additional data from our recent screening that contains snapshots and tomograms showing potential fusion events at the PM as depicted by the continuity of the CPV-II membrane with the inner leaflet of the PM. This data provides evidence albeit preliminary for fusion of CPV-II with the PM, setting the tone for further studies using cryo-ET that is currently in the works. The authors have also added their viewpoint about such egress systems considering what's known in other viruses with dual mode of egress. The authors thank the reviewer for his/her suggestion and urging us to do so.

2) The CPV-II are implied- including in the title - as originating from the Golgi only. In the discussion they conclude that "prevalent forms of CPV-II carried the Golgi HRP-marker ". These data only show that at least some of the membrane is from the Golgi, since the identity of a continuous membrane that is labelled by a protein in the bilayer may be a patchwork of contributions. The contribution from pre or post-Golgi organelles is not investigated here, nor discussed, despite two possible contributors; firstly the complex set of double-membrane structures thought to form from the ER and associated with replication, secondly the commonly seen contribution from endosomal membrane/proteins to secretory organelles - and the CPV is proposed to lead to viral release. Investigating how the ER or endosomal - derived membranes relate to the Golgi-derived structures seen here is not needed for this paper, but the relation of these structures needs at least to be discussed.

Response: The authors acknowledge the concerns posed by the reviewer regarding the title of the study that apparently pushes for the notion that Golgi apparatus is the sole organelle that contributes membrane towards the biogenesis of CPV-II without ruling out the contribution of other pre-and post-Golgi organelles such as the ER or the endosomal system. The authors acknowledge this could be misleading. Thus, the authors have revised the title to facilitate a more accurate description of the study

that follows. We have also described this notion in the revised discussion. However, authors would also like to clarify that the primary goal of the paper was to investigate the contribution of Golgi in the formation of CPV-II and most importantly ask a topological question as to how the Golgi is restructured from flat-sheet like cisternae to the various forms of the CPV-II system. A comprehensive 3D model of this structural morphogenesis was lacking despite the knowledge of the involvement of Golgi in the biogenesis of CPV-II since the 1980s. As a follow up on the findings in his paper, HRP and APEX2 tagged organelle markers are being optimized for a more comprehensive picture of membrane contribution from other organelles. The authors thank the reviewer for pointing out this apparent ambiguity of the proposed title.

3) The topology of the structures is complex, and the location and orientation of the spike proteins and how these viruses are ultimately released by fusion of such complex structures with the plasma membrane is not addressed in the Model of Fig 10, (where it would help the reader to keep track of how this critical aspect of viral formation relates to the membrane remodeling) nor in the text- perhaps not surprising since Fig 10 is not mentioned in the text.

Response: We thank the reviewer for pointing this out. We have revised this figure (now Figure 9) to more comprehensive and better reflect on the model to help the readers track how the critical aspects of viral formation/membrane remodeling may be occurring (Figure 9A now). Orientation of the envelope protein has now been shown in new Figure 9B to display a cartoon structure of CPV-II classes 2-4, the orientation of the E2 glycoprotein, and how there may be fusion and or delivery of NC by CPV-II to the plasma membrane.

4) In Fig. 10, many of the NCs seem likely to be trapped in a dead-end structure. Indeed, one interpretation of Fig. 10 would be that only 2 virions in that diagram are going to be secreted free of wrapping membrane and ready to re-infect. This would be an extraordinarily wasteful process! Some speculation regarding membrane remodeling in late infection would be appropriate, or at least a clear statement that we are only seeing an intermediate stage and that there is another whole set of remodeling(s) required before release- if that is the authors view.

Response: The authors thank the reviewer for critical question of efficiency of this pathway and the modeling of the pathway. In fact, our original figure 10 was not clearly well thought out and we have revised (Now Figure 9A) to more carefully reflect our previous and new data. We hypothesize that once the CPVII fuses, the luminal contents expelled into the extracellular space may contain fully mature virions as well NC clusters in various stages of budding. New data (Figure 8) added to this revised version show a large sac like structure connected to the exterior of the cell, where CPV-II seems to be emptying their contents, in which we find representatives of each of these possibilities indicating that it is not a perfect system. Whether these incomplete intermediates expelled out of the cell serves as infectious units is an open question and is being currently pursued.

Further comments:

1) Evidence for accumulation of CPVII at the plasma membrane could be stronger. In the example shown in Figure 1 E(i) it is difficult to make out the plasma membrane. A dotted line or arrows would be useful to denote the position of the plasma membrane; this should be added to Figure 2 as well. To corroborate the example shown in Figure 2 B (i and vi) further examples should be included in supplementary.

Response: Additional images and snap shots of segmented tomograms have been added in this revised version of the manuscript (Figure 6F and H). Here multiple clusters of CPV-II color coded for different classes of CPVII are clearly shown.

2) Figure 5 A and B - Particularly in panel A; using two shades of green makes it difficult to differentiate between the two structures. An alternative to the dark green should be used or the authors could consider adding further images showing the green and dark green structures alone so that the reader can see the position and differentiate between the two.

Response: The color of Golgi cisternae in Fig 5A and B (now Fig 4) has now been changed to magenta from dark green for better visibility. Thanks for the suggestion.

3) There is a reference in the text to Fig 7H (line 389). This is not present in the figure that I have in the merged file that I am working with, and lines 391/392 also mention 7f and other similarly named figures that are not present. Further, 393 has a sentence starting "First...." and never gets to second and maybe would have been all the way to tenth! lines 388 to 395 refer to panels in Figure 7 that do not exist, and it may be that this paragraph should be referring to Figure 8.

Response: The missing section of Figure 7 (now figure 6H) has now been added. The authors thank the reviewer for pointing out this inconsistency. This was a mistake in making the final tiff for original upload where the bottom figure was cut out.

4) Figure 3B - Could include XY, XZ and YZ labels in the panel for the different views of the tomogram. In the XZ view the Golgi that can be seen, is it part of stack 1 or 3? If so, include the label in the XZ window.

Response: The authors have added the axis labels and added a label for Stack 1 in the XZ view in Figure 3B (Now, Figure 2B).

5) Figure 4 B and F - Dotted box width does not match the regions shown in C, D, G and H. G and H shows a larger area and lower magnification than C and D, hence the small NC sphere size and therefore it would be beneficial to the reader to have scalebars in B-D and f-H.

Response: We have removed the dotted lines in Fig 4B and 4F (now figure 3B and 3F) and added scalebars to B-D and F-H as suggest by the reviewer.

6) Line 377 - Figure 4F should be Figure 7F.

Response: We have made the correction. Thanks for finding the error.

7) Should include a supplementary video of Golgi stack 1 and Golgi stack 2 showing slices from the tomogram transitioning into the model shown in Figure 4 A-B and E-F.

Response: Added movie as suggested.

8) Sometimes refer to Fig and other times Figure - see line 139, 198, 203, 278, 288, 331, 335, 629.

Response: This inconsistency was rectified.

Reviewer #3 (Comments to the Authors (Required)):

1) Alphaviruses have similar size and genome, but vary in their pathogenicity, preferred host receptors and cellular sites for virus genome transcription, packing of virions and budding sites. Therefore, these results cannot be regarded as a general mechanism for alphavirus infection and trafficking. Although I

admired the quality of the tomograms, conclusions reached too far as they were based only to a portion of pleomorphic CPV-II from one cell. Authors present a four-step pathway for CPV-II biogenesis, which is interesting, but too speculative. Moreover, manuscript was very heavy to read, as it lacked focus, contained too much details of work that could be considered as preparatory work, and was full of small writing errors. I consider this manuscript too immature and its scientific impact too low to be suitable for publication.

Response: First, the authors thank the reviewer for sharing her/his insights and comments on this specific group of viruses and the constructive criticism of this manuscript. We agree with the reviewer's comment that alphaviruses though, quite similar in their make-up of their genome varies widely in their pathogenicity and capability to cause disease. However, the aspect we targeted in our study was the cell biology of VEEV that may or may not be directly linked to its pathogenicity. The goal of this work is to nucleate the idea of how a virus changes the two critical aspects of a premier secretory organelle of the cell known to be critical for its function: its architecture and localization. This leads to the formation of mobile vesicular structures loaded with the two critical factors indispensable for budding right up to the site of egress. The interest in this work is centered around this cellular phenomenon that has been observed in most alphaviruses (Western equine encephalitis, eastern equine encephalitis virus, Sindbis virus, Semliki Forest virus and Chikungunya studies till date). In fact, unpublished work from our lab has already found identical intermediates in Chikungunya virus.

For many years CPV-II have been neglected for in-depths studies but recent work in the last decade (Soonsawad et al. 2010 PLoS Pathogens, Jose et al. 2017 mBio, Elmarsi et al 2021 Pathogens) has indicated that CPV-II may not be just a bystander and could be directly contributing to the late cycles of infections. This drastic change of the Golgi architecture must have significant effect on the secretory status during infection with alphavirus. The fact that the cell survives this secretory remodeling for days in some alphaviruses brings in the idea of a structurally remodeled secretory system that carries on the critical functions of the cell. This paper lays down the foundations for all those studies and ushers in the important idea on how the secretory system of the cell is reorganized by the virus. The functional studies on these structural rearrangements of the system will be critical to dissect this phenomenology. Thus, we respectfully disagree that the paper is low in impact and scientifically immature. Not just for the sheer amount of work and planning that was put in to understand this organelle remodeling but also for its larger implications in the field of the cell biology of this important group of viruses. The revised manuscript also includes new data to answer reviewer #1 and 2 queries. We have also cut down on the manuscript text and made it more focused. Similarly, we have added description that because alphaviruses differ in pathogenicity, there may be differences in CPV-II class formation/efficiency, etc.

Here are minor comments on how to improve the manuscript.

1) The manuscript could be made shorter and more focused throughout, for example by removing the comparisons to autophagosome formation in the introduction, results, and discussion, as there is no evidence to show that similar mechanism is used. In fact, there is lot of evidence of the opposite. Especially text in lines 407-417 is pure speculation and should not be in the results section.

Response: We thank the reviewer for her/his candid comments on our work and help to make this a better manuscript. In our effort to make the paper more concise we have rewritten several sections for better flow and have tried to make the manuscript more focused. For example, we have removed the topological comparison of one of the CPV-II forms and its maturation with that of autophagosomes from both result and discussion.

2) The beginning of the results contains too many results (e.g. screening of optimal MOI for virus

infection and TEM screen from different time points post infection) that could be considered as preparatory work that need not to be reported with such details. Figure 1 is not informative; especially I cannot see any point for Figure 1C, unless the aim is to show that these sections exist. Figure 1A shows Golgi cisternae without any contrast, in comparison to most of the other images. Is this a processing artefact? Moreover, if the structures are devoid of NCs (lanes 133-134), how one can be sure that this is from an infected cell? NCs should be visible in 90 nm thin sections, as seen in other images too. Cisternal bending depicted in purple in Figure 1Biii is quite typical for Golgi cisternae. Figure 1D shows collage of projection images from 250 nm thick section with very low informational value due to overlapping structures. I would seriously consider leaving the whole Figure 1 out.

Response: We acknowledge the fact that the Figure 1 in the older version of the manuscript was a little busy and could be deemed as preliminary data by some, as this did. However, our prior experience with another reviewer when we omitted this part, has been the opposite and we were asked to include this in our figure to show we have carried out extensive studies to make sure that the result we get from the representative cell is not a fluke that we happened to come across and decided to do a 3D study on. Therefore, taking this contrasting point of views into account, we have now moved Figure 1 to our supplementary section (Figure S1).

3) Figure 2 has very little membrane contrast, and trans cisternae cannot be seen at all. Is this in all cells? I have seen nicer morphology with standard embedding techniques.

Response: The authors thank the reviewer for her/his comments. As α mannosidase-II is a cis-medial marker, we did not really expect to detect the trans-Golgi. Unfortunately, after much optimization, we were unable to obtain the same efficacy of membrane contrast seen here with the FLIPPER construct (see Kuipers et al. 2015). However, since our goal was to find out if the different forms of CPV-II carry this marker, one that was confirmed, we did not go in for further optimization for better membrane staining. However, it should be noted that we were able to get better membrane preservation and contrast in HeLa cells expressing α ManII-HRP (please see Fig S2) using the same protocol.

4) lane 131: ...typical stacked architecture.. is depicted with white arrow (not yellow as in the text)

Response: This error has been rectified. Thank you.

5) lane 192: TC-83 is mentioned here, and explained only in the discussion part

Response: This discussion of TC-83 has now been moved to the results section. Thank you.

6) lane 198: instead of osmium stain, should it be DAB precipitate?

Response: Thanks for this suggestion. We write osmium stain as the DAB is osmophilic and it's the preferential staining of DAB by osmium that is detected over the non-specific staining of other cellular membranes.

7) lanes 293-295: This part is very speculative, if there was there only one intermediate structure of this type

Response: Such intermediates extremely abundant in our data set and formed the basis of our hypothesis. However, this example was that of the largest of them and was easy to demonstrate the components of both a flat cisterna as well as vessel. Thus, we used this one single structure as an

example of the pivotal intermediate. The authors apologize for unintentionally using language that confused the reviewer.

lane 389: I could not find Figure 7H.

Response: Figure 7H has now been added. We apologize for the error as it was cut off in our final tiff preparation in the original submission.

lane 393: ... two common features... What is the second one?

Response: We have taken care of this incomplete sentence. Thanks for pointing it out.

lanes 391-393: not clear which images are referred to

Response: The images being referred to were from the missing 7H. This figure has now been added in the revised version and is now referred to as figure 6H.

lane 402-402: comparison to Figures 4B and 4C does not make any sense. Should it be Figures 7B and 7C?

Response: This section has been re-written to make it more concise and accurate.

lane 436: NC in green shown with an arrow is in Figure 9O

Response: This has been rectified.

lane 503: lipid bilayer thickness should be 4 nm and not microns

Response: We have rectified this typo and apologize for the error.

Abbreviation to post-infection (PI) is introduced on line 120, and then not used over 20 times after that. Similar inconsistencies with the use of many other abbreviations too.

Response: We have tried our best to take care of these inconsistencies in our resubmission.

September 6, 2022

RE: Life Science Alliance Manuscript #LSA-2020-00887-TR

Prof. Robert V Stahelin
Purdue University West Lafayette
Medicinal Chemistry and Molecular Pharmacology
207 S. Martin Jischke Drive
Room 446 DLR Building
West Lafayette, IN 47907

Dear Dr. Stahelin,

Thank you for submitting your revised manuscript entitled "Contribution of the Golgi apparatus in morphogenesis of a virus induced cytopathic vacuolar system". We would be happy to publish your paper in Life Science Alliance pending final revisions necessary to meet our formatting guidelines.

- please add ORCID ID for secondary corresponding author-they should have received instructions on how to do so
- please add a category for your manuscript to our system
- please use the [10 author names, et al.] format in your references (i.e. limit the author names to the first 10)
- we encourage you to introduce your panels in your figure legends in alphabetical order
- please add figure callouts for Figure S3 C,D; Figure S5 B-D; Figure S6 A-C (you currently have a figure callout for Figure S6E, but this is not in the figure legend or in the figure); Figure S7 A-D; Figure S9 A-B

Figure Check:

- Figure S3: the yellow boxes can be placed more precisely to indicate the area in the magnified version on the right. please mention the yellow boxes in the figure legend

A. FINAL FILES:

B. MANUSCRIPT ORGANIZATION AND FORMATTING:

Sincerely,

Reviewer #1 (Comments to the Authors (Required)):

The Authors seriously addressed all my comments and improved the manuscript over the original version. I advise to accept the manuscript for publication.

-please add ORCID ID for secondary corresponding author-they should have received instructions on how to do so.

Response: Thank you. The first author (secondary corresponding author) is linking their ORCID ID with the submission.

-please add a category for your manuscript to our system.

Response: Three subject categories have been added: Cell biology, molecular biology and microbiology, virology and host pathogen interactions.

-please use the [10 author names, et al.] format in your references (i.e. limit the author names to the first 10).

Response: The correction has been made for references with more than 10 authors.

-we encourage you to introduce your panels in your figure legends in alphabetical order.

Response: We have done our best to have figure legends addressed in alphabetical order.

-please add figure callouts for Figure S3 C,D; Figure S5 B-D; Figure S6 A-C (you currently have a figure callout for Figure S6E, but this is not in the figure legend or in the figure); Figure S7 A-D; Figure S9 A-B.

Response: We have made the appropriate callouts in the text to the different figure panels. Figure S6E was a typo and has been corrected to Figure S5D.

Figure Check:

-Figure S3: the yellow boxes can be placed more precisely to indicate the area in the magnified version on the right. please mention the yellow boxes in the figure legend.

Response: We thank the editors for noting this. The correct positioning of the yellow box has been made as well as the description in the figure legend.

Reviewer #1 (Comments to the Authors (Required)):

The Authors seriously addressed all my comments and improved the manuscript over the original version. I advise to accept the manuscript for publication.

Response: We thank the reviewer for reading our response to reviewers' comments and revised manuscript.

September 8, 2022

RE: Life Science Alliance Manuscript #LSA-2020-00887-TRR

Prof. Robert V Stahelin
Purdue University West Lafayette
Medicinal Chemistry and Molecular Pharmacology
207 S. Martin Jischke Drive
Room 446 DLR Building
West Lafayette, IN 47907

Dear Dr. Stahelin,

Thank you for submitting your Research Article entitled "Contribution of the Golgi apparatus in morphogenesis of a virus induced cytopathic vacuolar system". It is a pleasure to let you know that your manuscript is now accepted for publication in Life Science Alliance. Congratulations on this interesting work.

DISTRIBUTION OF MATERIALS:

Again, congratulations on a very nice paper. I hope you found the review process to be constructive and are pleased with how the manuscript was handled editorially. We look forward to future exciting submissions from your lab.

Sincerely,
